# Cortical direction selectivity increases from the input to the output layers of visual cortex

**Weifeng Dai[1], Tian Wang[1,2], Yang Li[1], Yi Yang[1], Yange Zhang[1], Yujie Wu[1], Tingting Zhou[1], Hongbo Yu[3], Liang Li[4], Yizheng Wang[4], Gang Wang[4]\*, Dajun Xing[1]\***

**1** State Key Laboratory of Cognitive Neuroscience and Learning & IDG/McGovern Institute for Brain Research, Beijing Normal University, Beijing, China, **2** College of Life Sciences, Beijing Normal University, Beijing, China, **3** School of Life Sciences, State Key Laboratory of Medical Neurobiology, Collaborative Innovation Center for Brain Science, Fudan University, Shanghai, China, **4** Beijing Institute of Basic Medical Sciences, Beijing, China

\* g_wang@foxmail.com (GW); dajun_xing@bnu.edu.cn (DX)

**Data Availability Statement:** All relevant data are within the paper and its Supporting Information files. All data files and analysis code are available from the Zenodo database (URLs: https://doi.org/10.5281/zenodo.14177191).

## Abstract

Sensitivity to motion direction is a feature of visual neurons that is essential for motion perception. Recent studies have suggested that direction selectivity is re-established at multiple stages throughout the visual hierarchy, which contradicts the traditional assumption that direction selectivity in later stages largely derives from that in earlier stages. By recording laminar responses in areas 17 and 18 of anesthetized cats of both sexes, we aimed to understand how direction selectivity is processed and relayed across 2 successive stages: the input layers and the output layers within the early visual cortices. We found a strong relationship between the strength of direction selectivity in the output layers and the input layers, as well as the preservation of preferred directions across the input and output layers. Moreover, direction selectivity was enhanced in the output layers compared to the input layers, with the response strength maintained in the preferred direction but reduced in other directions and under blank stimuli. We identified a direction-tuned gain mechanism for interlaminar signal transmission, which likely originated from both feedforward connections across the input and output layers and recurrent connections within the output layers. This direction-tuned gain, coupled with nonlinearity, contributed to the enhanced direction selectivity in the output layers. Our findings suggest that direction selectivity in later cortical stages partially inherits characteristics from earlier cortical stages and is further refined by intracortical connections.

## Introduction

As one of the most fundamental visual attributes, motion information is essential for visual information processing. Accordingly, neurons within the visual system of various species exhibit selectivity for motion direction (cat: [1]; monkey: [2]; ferret: [3]; rabbit: [4]; mouse: [5]; fly: [6]). Despite extensive research into the generation of direction selectivity at the initial stage [7–14], questions remain about how direction selectivity is processed and relayed at subsequent stages.

**Funding:** This study was supported by the National Natural Science Foundation of China (https://www.nsfc.gov.cn/english/site_1/index.html, 32171033 to D.X. and 32100831 to T.W.), the Ministry of Science and Technology of the People's Republic of China (https://en.most.gov.cn/) STI2030-Major Projects (2022ZD0204600 to D.X.), and the Fundamental Research Funds for the Central Universities to D.X. (http://en.moe.gov.cn/). The funders had no role in study design, data collection and analysis, decision to publish, or preparation of the manuscript.

**Competing interests:** The authors have declared that no competing interests exist.

**Abbreviations:** AIC, Akaike information criterion; BIC, Bayesian information criterion; CSD, current source density; DSI, direction selectivity index; GC, Granger causality; HWHH, half-width at the half-height; LFP, local field potential; MT, middle temporal; MUA, Multiunit spiking activity; OSI, orientation selectivity index; PSTH, peristimulus time histogram; RF, receptive field; SNR, signal-to-noise ratio; SPN, sparse noise; SU, single unit; SUA, single-unit spiking activity.

It is believed that direction selectivity in later cortical stages substantially inherits characteristics from earlier cortical stages [15], as observed for the laminar processing of orientation selectivity [16–19], black–white asymmetry [20–23], and surround modulation [24–27] in the primary visual cortex (V1). Under this hypothesis, the direction selectivity of later cortical stages should strongly correlate with earlier cortical stages, sharing similar preferred directions. Consistent with this hypothesis, direction-selective cells in V1 are likely to receive synaptic connections from other V1 cells that have similar preferred directions [28–33]. In contrast, recent studies have suggested that direction selectivity is re-established at multiple stages along the visual pathway. In mice, direction selectivity first occurs in the retina [34] and is then independently reestablished in L4 [9] and L2/3 [35] of V1. Furthermore, the preferred direction of postsynaptic neurons in L2/3 of mouse V1 does not correlate with that of presynaptic neurons in L4 [35,36]. A similar de novo generation of direction selectivity at later stages has been observed in L4 of V1 in rabbits [37,38] and in the middle temporal (MT) visual area of monkeys [39]. Given these 2 contrasting hypotheses, the inheritance hypothesis and the re-establishment hypothesis, the manner in which direction selectivity is processed and relayed through consecutive cortical stages of the visual hierarchy, especially within the neocortex's laminar structure [40], remains to be fully elucidated.

Leveraging the intricate neural circuitry within a cortical column [16,41–43] and the advancements in laminar recording [44–46], we investigated the processing and relay of direction selectivity across 2 successive stages of the visual hierarchy in cats: the input layers and the output layers within the early visual cortices, including areas 17 and 18. Unlike in monkey V1 [47], where direction-selective neurons are prevalent only in the input layers (layers 4 and 6), cat early visual cortices have direction-selective neurons within both the input and output layers [48–50]. This enables us to investigate the laminar transmission of direction selectivity and how laminar circuits process direction selectivity.

By employing drifting grating stimuli with varied directions, we assessed alterations in the strength of direction selectivity and shifts in the preferred direction as neural signals progressed from the input layers to the output layers within a single cortical column. Our findings indicate that the output layers not only inherit direction selectivity from the input layers but also exhibit enhanced direction selectivity. Through comparative analyses of population-averaged direction tuning curves and the development of computational models, we discovered a direction-tuned gain mechanism for signal transmission from the input layers to the output layers, which leads to enhanced direction selectivity in the output layers. Granger causality (GC) analysis suggested that this direction-tuned gain mechanism may be the result of both feedforward connections from the input layers to the output layers and recurrent connections within the output layers.

## Results

In the present study, we recorded laminar responses in areas 17 and 18 of anesthetized cats with multielectrode linear probes (**Fig 1A**). Multiunit spiking activity (MUA), representing the response of a relatively local population [51], as well as sorted single-unit spiking activity (SUA), were extracted for each recording site. Unless otherwise specified, results from MUA were shown. We verified the perpendicularity of each probe placement by confirming minimal variation in RF center positions and high consistency of preferred orientations across recording sites (**S1 Fig**). Additionally, we determined the laminar position of each recording site based on the MUA, local field potential (LFP), and current source density (CSD) responses to sparse noise (SPN) stimuli (**S2 Fig**).

Based on neuronal responses to SPN stimuli within the receptive field (RF) center, we calculated the response latency of each recording site, defined as the time point when the

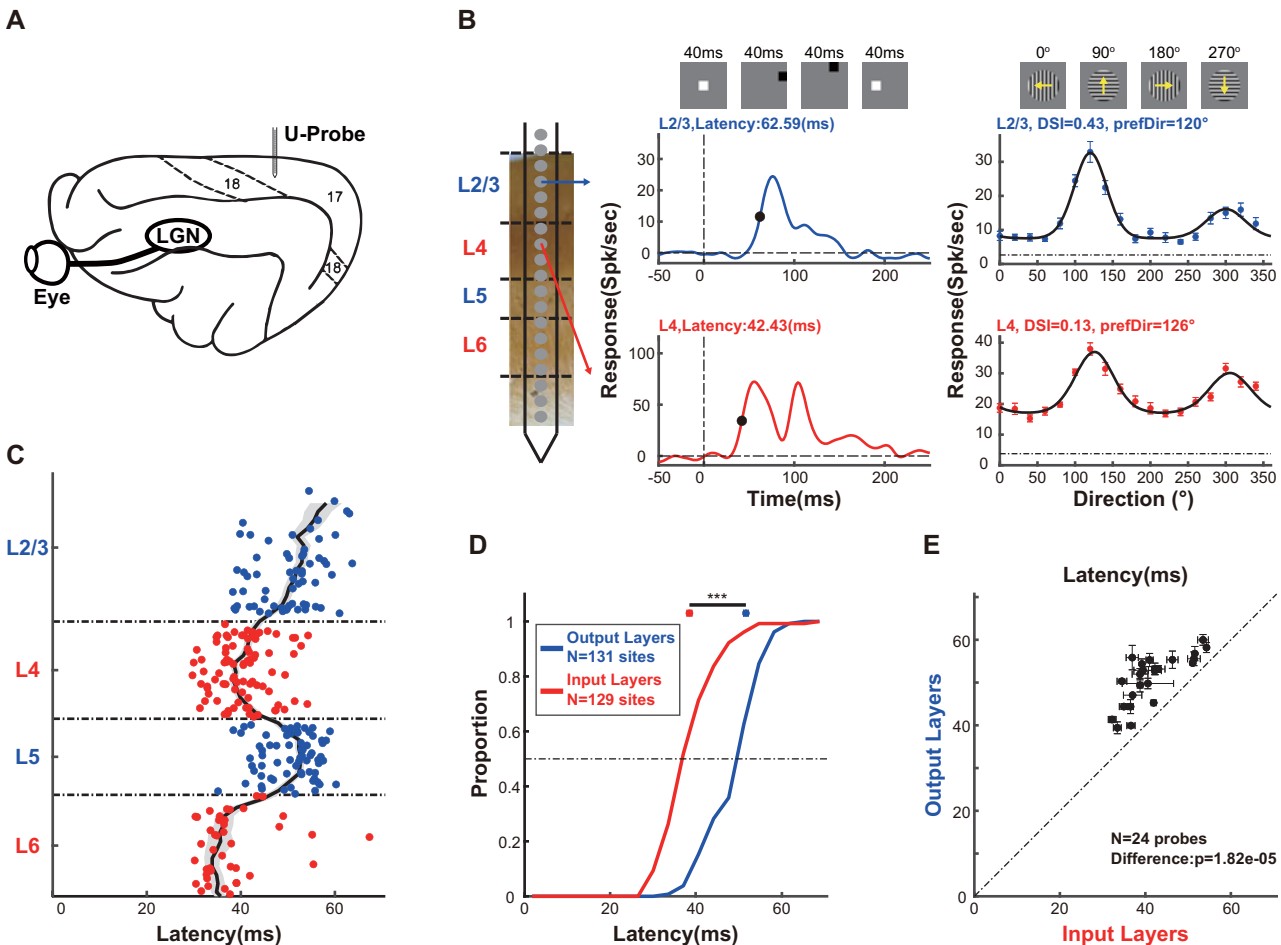

**Fig 1. Neuronal responses are delayed in the output layers.** **(A)** Basic methods. Utilizing multielectrode linear probes, we recorded laminar responses in areas 17 and 18 of anesthetized cats. Unless otherwise noted, MUA results were shown in this and other figures. **(B)** Neuronal responses of example recording sites from L2/3 (upper) and L4 (lower). Left, a linear probe superimposed on a cortical section stained with CO. Middle, averaged neuronal responses (blue and red lines) to SPN stimuli located within the RF center. We defined the neuronal response latency (black dots) as the time point at which the response reaches half of its maximum response. The vertical dashed line indicates the stimulus onset time, and the horizontal dashed line represents 0 spk/sec, as the neuronal response was subtracted from the baseline response before stimulus onset. Right, neuronal responses in different directions (blue and red dots), overlaid with fitted direction tuning curves (black solid line) using the von Mises function. Data are mean ± SEMs. The preferred direction and the DSI were extracted for each recording site. The DSI was calculated as $\frac{R(prefDir)-R(nullDir)}{R(prefDir)+R(nullDir)}$, where $R(prefDir)$ and $R(nullDir)$ are the neuronal responses in the preferred direction and in the null direction (the direction opposite to the preferred direction), respectively. The horizontal dashed line indicates the neuronal response to blank stimuli. **(C)** Laminar pattern of response latency. Each dot represents the data from one recording site. Recording sites belonging to the output layers, including L2/3 and L5, are shown in blue. Recording sites in the input layers, including L4 and L6, are shown in red. The running medians of neighboring recording sites are shown as black lines, and shaded error bars indicate the SEM. **(D)** Cumulative distributions of response latency in the output layers (blue line) and the input layers (red line). The horizontal dashed line indicates a cumulative proportion of 0.5. The median value and SEM of response latency for input and output layers are shown at the top. Statistical comparison for median response latency between the output layers and the input layers is shown at the top. **(E)** Comparison of response latency between the output layers and the input layers within the same probe placement. For each probe placement, we obtained the median value and SEM of response latency across all recording sites in the output layers and in the input layers. Thus, each dot represents data from 1 probe placement. The significance of paired comparisons between the output layers and the input layers is shown. Data are median ± SEMs. ns, not significant, *$p < 0.05$, **$p < 0.01$, ***$p < 0.001$. Data and code that support these findings are available at: https://doi.org/10.5281/zenodo.14177191. CO, cytochrome oxidase; DSI, direction selectivity index; MUA, Multiunit spiking activity; RF, receptive field; SPN, sparse noise.

neuronal response reaches half of its maximum (**Fig 1B**). By pooling all the recording sites of each layer, we characterized the laminar distribution of response latency (**Fig 1C**). When comparing the input layers (including L4 and L6) with the output layers (comprising L2/3 and L5), we found that neuronal responses are significantly delayed in the output layers (**Fig 1D**; output

layers: 51.76±0.5479 ms, $N$ = 131 recording sites; input layers: 38.55±0.5716 ms, $N$ = 129 recording sites; interlaminar difference: $p$ = 3.6405*10$^{-26}$; Wilcoxon rank-sum test), and this delay exists in all probe placements (**Fig 1E**, $N$ = 24 probe placements, $p$ = 1.8215*10$^{-5}$; Wilcoxon signed-rank test). This latency variation across cortical depth justified the laminar assignment within the early visual cortices in our study.

## Direction selectivity is enhanced in the output layers

Based on the neuronal responses to drifting gratings with varied directions, we obtained direction tuning of each recording site, from which we calculated the direction selectivity index (DSI), which represents the strength of direction selectivity, and determined the preferred direction, which is the direction that evokes maximum responses (**Fig 1B**).

We then characterized the laminar distribution of the DSI (**Fig 2A**). We observed that the output layers exhibited a significantly larger DSI (**Fig 2B**; output layers: 0.4605±0.0161, $N$ = 134 recording sites; input layers: 0.2566±0.0194, $N$ = 109 recording sites; interlaminar difference: $p$ = 1.7392*10$^{-8}$; Wilcoxon rank-sum test) and had a larger proportion of recording sites with strong direction selectivity (**Fig 2B**; proportion of recording sites with DSI > 1/3, output layers: 73.13%, $N$ = 134 recording sites; input layers: 43.12%, $N$ = 109 recording sites; interlaminar difference: $p$ = 2.0992*10$^{-6}$; chi-square goodness-of-fit test), indicating an enhancement of direction selectivity in the output layers. This enhancement was found in almost all probe placements (**Fig 2C**, $N$ = 25 probe placements, $p$ = 2.5432*10$^{-5}$; Wilcoxon signed-rank test). Across all probe placements, we found that the DSIs in the output layers were approximately 78.59% greater than those in the input layers (**Fig 2D**, DSI enhancement: 0.7859±0.1813, $N$ = 25 probe placements). Additionally, our analysis revealed a strong positive correlation between the DSIs in the output layers and those in the input layers (**Fig 2C**, $N$ = 25 probe placements, $r$ = 0.7131, $p$ = 9.7213*10$^{-6}$; Spearman's $\rho$), suggesting that the direction selectivity observed in the output layers may be partially inherited from the input layers. Similar results were obtained when considering only L2/3 and L4 (**S3A–S3C Fig**).

In addition to comparing DSIs, we quantified the shifts in preferred directions of neuronal pairs located at different depths within the output layers, within the input layers or across the input and output layers. We observed that for most neuronal pairs within a cortical column, the preferred direction remained largely unchanged, irrespective of whether the neurons were within the output layers, within the input layers or across the input and output layers (**Fig 3A**, proportion of neuronal pairs with preferred direction difference less than 30 degrees, within output layers: 83.33%, $N$ = 330 pairs; within input layers: 69.62%, $N$ = 237 pairs; across input and output layers: 78.16%, $N$ = 577 pairs), reflecting the columnar organization of direction preference previously demonstrated in the early visual cortices of carnivores (cat: [52–54]; ferret: [3,55]). We also found a small subset of neuronal pairs exhibiting a shift in preferred directions of approximately 180 degrees (**Fig 3A**, proportion of neuronal pairs with preferred direction difference greater than 150 degrees, within output layers: 10.30%, $N$ = 330 pairs; within input layers: 20.25%, $N$ = 237 pairs; across input and output layers: 14.38%, $N$ = 577 pairs), consistent with findings from previous studies [56]. However, upon comparing neuronal pairs with strong direction selectivity (those for which the DSIs of both recording sites were greater than 1/3), we found that nearly all pairs exhibited no change in their preferred directions (**Fig 3B**, proportion of neuronal pairs with preferred direction difference smaller than 30 degrees, within output layers: 86.74%, $N$ = 181 pairs; within input layers: 88.41%, $N$ = 69 pairs; across input and output layers: 88.99%, $N$ = 218 pairs). Enhancement of direction selectivity in the output layers and maintenance of preferred directions across input and

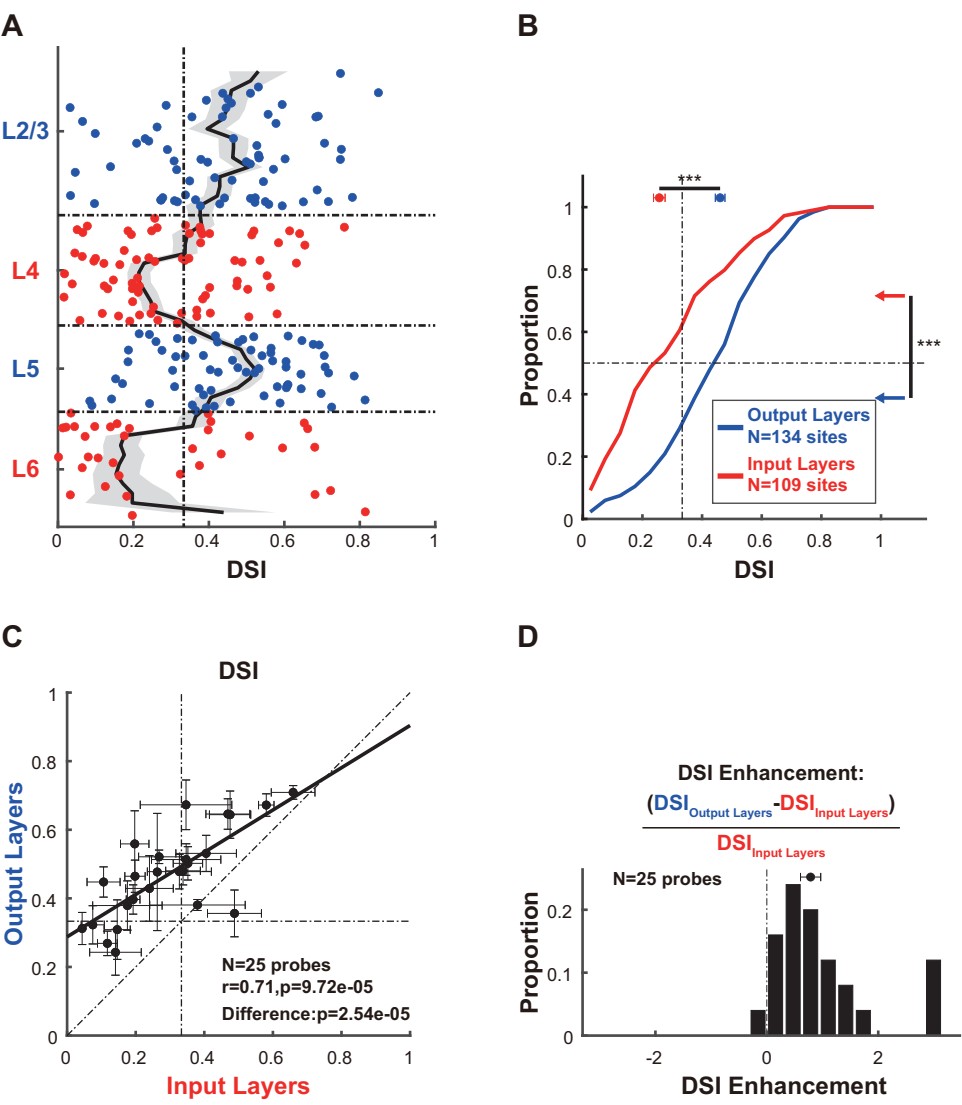

**Fig 2. Direction selectivity is enhanced in the output layers. (A)** Laminar pattern of the DSI. Each dot represents the data from one recording site. Recording sites belonging to the output layers, including L2/3 and L5, are shown in blue. Recording sites in the input layers, including L4 and L6, are shown in red. The vertical dashed line represents a DSI of 1/3, indicating that the response strength in the preferred direction is twice that in the null direction. The running medians of neighboring recording sites are shown as black lines, and shaded error bars indicate the SEM. **(B)** Cumulative distributions of DSIs in the output layers (blue line) and the input layers (red line). The horizontal dashed line indicates a cumulative proportion of 0.5, and the vertical dashed line represents a DSI of 1/3. The median value and SEM of the DSI for both layers are shown at the top. The proportions of recording sites exhibiting strong direction selectivity (DSI > 1/3) within the input and output layers are denoted by colored arrows on the right. Statistical comparisons for median DSI values (at the top) and proportions of recording sites with strong direction selectivity (on the right) between the output layers and the input layers are also shown. **(C)** Comparison of DSIs between the output layers and the input layers within the same probe placement. For each probe placement, we obtained the median value and SEM of the DSI across all recording sites in the output layers and in the input layers. Thus, each dot represents data from one probe placement. The Spearman correlation and the significance of paired comparisons between the output layers and the input layers are shown. Data are median ± SEMs. **(D)** The distribution of DSI enhancements between the output layers and the input layers. The DSI enhancement was calculated as the ratio of the DSI difference between the output and input layers to the DSI in the input layers. Data and code that support these findings are available at: https://doi.org/10.5281/zenodo.14177191. DSI, direction selectivity index.

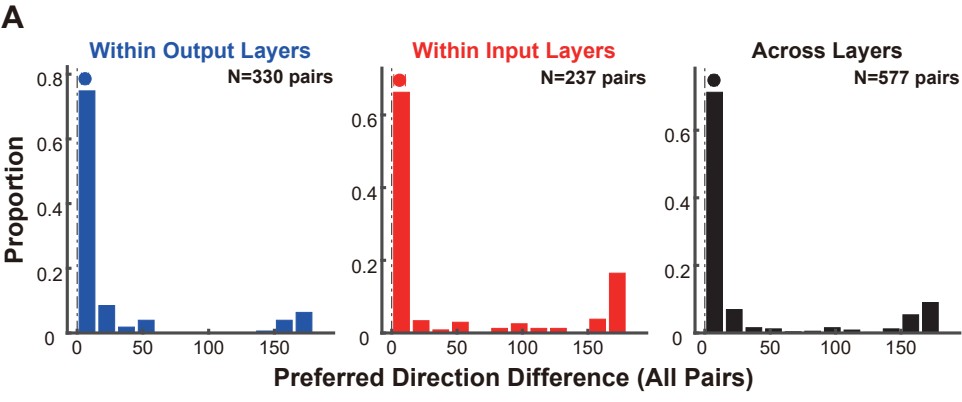

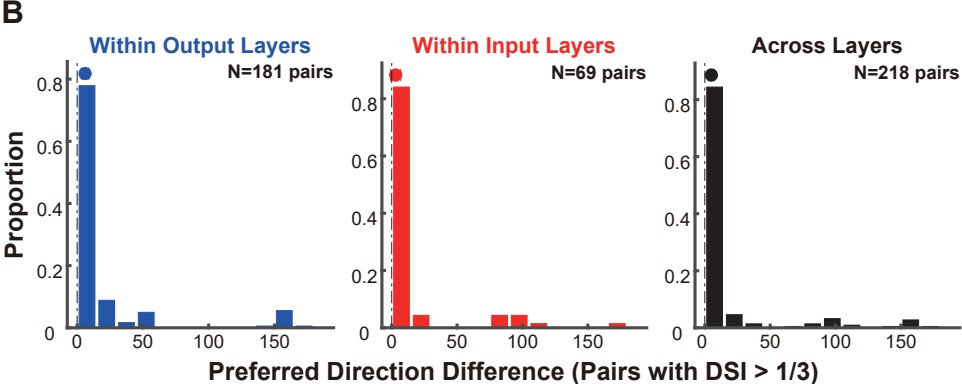

**Fig 3. Preferred directions are maintained within a cortical column. (A)** Differences in the preferred directions between all pairs of recording sites within the output layers (left panel), within the input layers (middle panel), and across the input and output layers (right panel). **(B)** Similar to A but restricted to pairs in which the DSIs of both recording sites were larger than 1/3. Data and code that support these findings are available at: https://doi.org/10.5281/zenodo.14177191. DSI, direction selectivity index.

output layers were also observed with sorted single units and when considering only putative excitatory neurons based on their spike waveforms (**S4A–S4D Fig**).

Through an interlaminar comparison of the DSIs, we demonstrated an enhancement in direction selectivity within the output layers of cat early visual cortices. The strong positive correlation between the DSIs of the output layers and the input layers, coupled with the consistent preferred directions across these layers, suggests that the direction selectivity observed in the output layers is predominantly inherited from the input layers.

## The response in the preferred direction is selectively preserved from the input layers to the output layers

To further elucidate the processing and relay of direction selectivity from the input layers to the output layers, we compared the population-averaged direction tuning between the input and output layers. Prior to averaging, we aligned the preferred direction of each recording site to 90 degrees. As depicted in **Fig 4A**, as opposed to the neuronal responses in the preferred direction, the response strength in the other directions was weakened in the output layers compared to that in the input layers. As a control, we also compared the response strength under blank stimuli between the output layers and the input layers. We found that, under blank stimuli, the response strength was also reduced in the output layers (**Fig 4A**), which is in

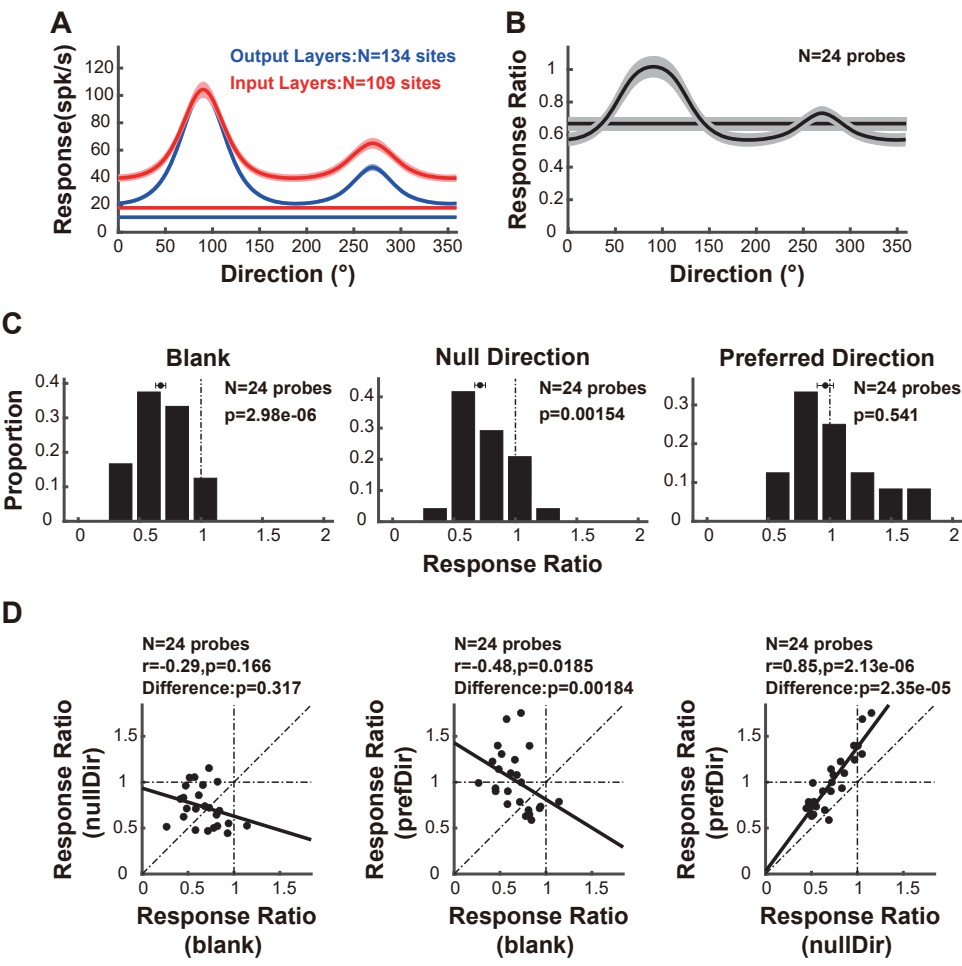

**Fig 4. The response in the preferred direction is selectively preserved from the input layers to the output layers.**
**(A)** Population-averaged direction tuning of response strength within the output layers (blue lines) and the input layers (red lines). The blue and red horizontal lines represent the average responses to blank stimuli within the output layers and within the input layers, respectively. Data are mean ± SEMs. **(B)** Population-averaged direction tuning of the response ratio between the output layers and the input layers. Data are mean ± SEMs. **(C)** Distribution of the response ratio under the indicated stimulus conditions. The vertical dashed line represents a response ratio value of 1. From left to right, the panels show the response ratio under blank stimulus, in the null direction, and in the preferred direction. **(D)** Comparison of the response ratios between the indicated stimulus condition pairs. From left to right, the panels depict the comparison of response ratios between the null direction and blank stimulus, between the preferred direction and blank stimulus, and between the preferred direction and the null direction. Each dot indicates the data from one probe placement. Data and code that support these findings are available at: https://doi.org/10.5281/zenodo.14177191.

line with the reduced spontaneous activity in the output layers of awake macaques [57–60] and suggests that neuronal activity in the output layers is inherently weaker than that in the input layers.

We then calculated the response ratio, which is the ratio of the response strength between the output layers and the input layers. The response ratio was greatest in the preferred direction (**Fig 4B**). **Fig 4C** shows the distribution of response ratios under 3 stimulus conditions: blank stimulus, null direction, and preferred direction. For both the blank stimuli and null direction, the response ratio was significantly less than 1 (**Fig 4C**; blank stimuli: $N = 24$ probe placements, $p = 2.98*10^{-6}$; null direction: $N = 24$ probe placements, $p = 0.00154$; sign test), indicating a less active neuronal response in the output layers under nonpreferred conditions.

In contrast, in the preferred direction, the response ratio approached 1 (**Fig 4C**, $N$ = 24 probe placements, $p$ = 0.541; sign test), suggesting the preservation of the preferred response in the output layers. A comparison of the response ratios among these conditions revealed no significant difference between the blank stimuli and null direction (**Fig 4D**, $N$ = 24 probe placements, $p$ = 0.317; Wilcoxon signed-rank test), whereas the response ratio for the preferred direction was substantially greater than that for both the blank stimuli (**Fig 4D**, $N$ = 24 probe placements, $p$ = 0.00184; Wilcoxon signed-rank test) and null direction (**Fig 4D**, $N$ = 24 probe placements, $p$ = $2.35*10^{-5}$; Wilcoxon signed-rank test). Additionally, there are diverse relationships between response ratios across different stimulus conditions (**Fig 4D**; null direction versus blank stimuli: $r$ = −0.29, $p$ = 0.166; preferred direction versus blank stimuli: $r$ = −0.48, $p$ = 0.0185; preferred direction versus null direction: $r$ = 0.85, $p$ = $2.13*10^{-6}$; Spearman's $\rho$). The interlaminar comparisons of response strength at these chosen stimulus conditions for each probe placement are shown in **S5 Fig**. Similar results were obtained when considering only the signal transmission from L4 to L2/3 (**S3D–S3G Fig**).

To summarize, the results indicate that neuronal responses are inherently less active in the output layers, a trend that persists for the nonpreferred directions. However, neuronal responses in the preferred direction are selectively preserved from the input layers to the output layers.

## A direction-tuned gain mechanism for interlaminar signal transmission leads to enhanced direction selectivity in the output layers

What mechanisms can lead to a direction-tuned response ratio between the output layers and the input layers and ultimately result in enhanced direction selectivity in the output layers? Similar to the neural circuits responsible for generating direction selectivity [10,61], the direction-tuned response ratio between the output layers and the input layers we observed could result from several factors: (1) Nonlinearity [50,62], such as spiking nonlinearity [63] or gating of sensory information across layers [64]. Under this assumption, a direction-independent gain of signal transmission from the input to output layers, combined with a nonlinearity like a threshold-linear function, is sufficient to explain the observed direction-tuned response ratio. (2) A direction-dependent change in the relative strength of excitation and inhibition, which could involve increased excitation in the preferred direction [13], increased inhibition in the null direction [14], or both, as seen with the influence of brain state on direction selectivity in rabbit LGN direction-selective neurons [38]. This mechanism suggests that, in contrast to a direction-independent gain, the gain of interlaminar signal transmission varies with the direction of the stimulus.

To differentiate between these 2 potential mechanisms, we constructed 2 models simulating signal transmission from the input layers to the output layers (**Fig 5A**). In both models, the neuronal response in the input layers was first scaled by a linear gain and then subjected to static nonlinearity, specifically a threshold-linear transfer function [65–67]. The parameters of both models were optimized to explain neuronal responses in the output layers. The distinction between these 2 models lies in the linear gain: in the first model, the gain was constant across all stimulus conditions (Model I: untuned gain), whereas in the second model, the gain varied with stimulus condition (Model II: tuned gain).

We first evaluated whether nonlinearity alone could explain the neuronal responses observed in the output layers. Although the DSI of the output responses in Model I exceeded that of the input layers (**Fig 5B**, $N$ = 24 probe placements, $p$ = 0.000228; Wilcoxon signed-rank test), it remained lower than that of the output layers (**Fig 5C**, $N$ = 24 probe placements, $p$ = 0.000162; Wilcoxon signed-rank test). The overall contribution of nonlinearity to the

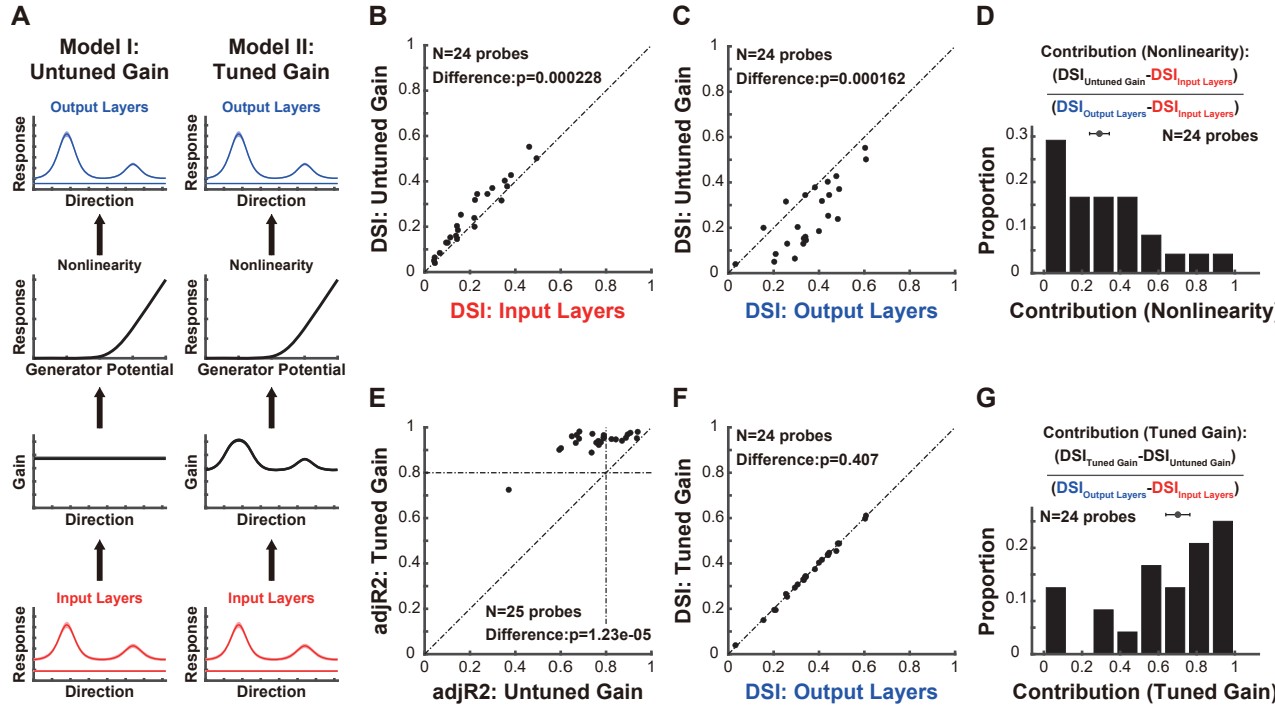

**Fig 5. A model with an interlaminar direction-tuned gain mechanism can explain neuronal responses in the output layers. (A)** Two models aim to explain neuronal responses in the output layers. In both models, responses from the input layers are first scaled by a linear gain and then pass through a static nonlinearity before producing the output responses. In the first model (Model I, Untuned Gain), the linear gain is fixed across all directions. In the second model (Model II, Tuned Gain), the linear gain varies in different directions. The exemplar neuronal responses in the input and output layers are the same as those in **Fig 4A**, and are also presented as mean ± SEMs. **(B)** Comparison of DSIs between the output of Model I and the input layers. **(C)** Comparison of DSIs between the output of Model I and the output layers. **(D)** The distribution of the contribution of nonlinearity to the enhancement of direction selectivity across layers. This contribution is quantified by calculating the ratio of the DSI difference between Model I (untuned gain) and the input layers to the DSI difference between the output layers and the input layers. **(E)** Comparison of adjusted goodness of fit ($adjR^2$) between Model I and Model II. Since Model I and Model II differ in the number of free parameters, we used $adjR^2$, which accounts for the number of free parameters in each model, to enable fair comparisons. The horizontal and vertical dashed lines represent an $adjR^2$ value of 0.8. Only results from probe placements where Model II has an $adjR^2$ greater than 0.8 were shown in other figures. Similar results were obtained using the AIC and BIC. **(F)** Comparison of DSIs between the output of Model II and the output layers. **(G)** The distribution of the contribution of the direction-tuned gain to the enhancement of direction selectivity across layers. This contribution is quantified by calculating the ratio of the DSI difference between Model II (tuned gain) and Model I (untuned gain) to the DSI difference between the output layers and the input layers. For B, C, E, and F, each dot represents data from one probe placement. Data and code that support these findings are available at: https://doi.org/10.5281/zenodo.14177191. AIC, Akaike information criterion; BIC, Bayesian information criterion; DSI, direction selectivity index.

enhancement of direction selectivity is about 29.09%±5.17% (**Fig 5D**). This finding indicates that while nonlinearity contributes to enhancing direction selectivity beyond the input layers, it does not entirely account for the enhanced direction selectivity observed in the output layers. In contrast, Model II offers a more comprehensive explanation of neuronal responses in the output layers, as indicated by the adjusted goodness of fit ($adjR^2$), which accounts for the number of free parameters in each model (**Fig 5E**, $adjR^2$ of Model II versus $adjR^2$ of Model I: $N = 25$ probe placements, $p = 1.23*10^{-5}$; Wilcoxon signed-rank test). Based on the Akaike information criterion (AIC) and Bayesian information criterion (BIC), which both account for the number of free parameters in each model, we obtained similar results (**S6 Fig**). The DSI of the output response in Model II shows no significant difference compared to that in the output layers (**Fig 5F**, DSI of the output responses in Model II versus DSI of the output layers: $N = 24$ probe placements, $p = 0.407$; Wilcoxon signed-rank test). The diverse relationships of response ratios between different stimulus conditions, as demonstrated in **Fig 4D**, can also be accurately replicated in Model II (**S7 Fig**). Additionally, the contribution of direction-tuned gain to the

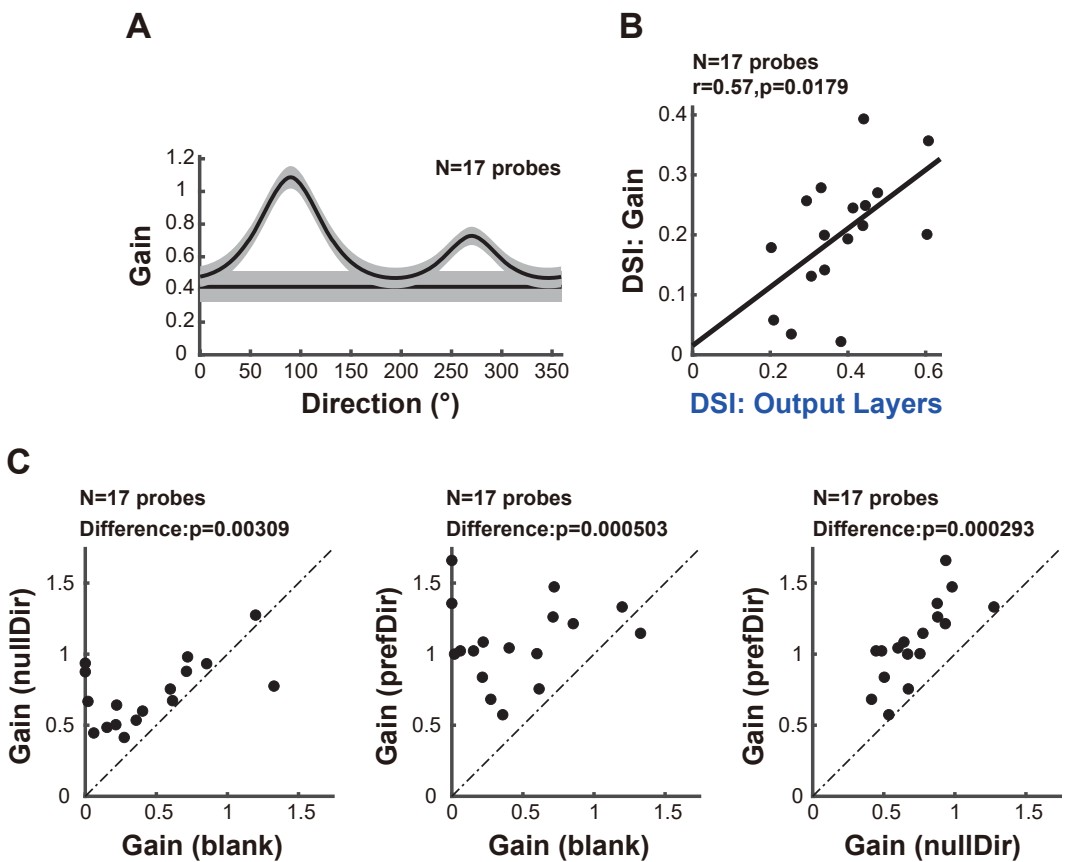

**Fig 6. A direction-tuned gain contributes to the enhancement of direction selectivity in the output layers. (A)**
Population-averaged direction tuning of linear gains in Model II. The horizontal line is the linear gain under the blank
stimulus. Data are mean ± SEMs. **(B)** Relationship of DSIs between the linear gain and the output layers. **(C)** Comparison of
linear gains between the indicated stimulus condition pairs. From left to right, the panels depict the comparison of linear
gains between the null direction and the blank stimulus, between the preferred direction and the blank stimulus, and between
the preferred direction and the null direction. For B and C, each dot represents data from one probe placement. Data and
code that support these findings are available at: https://doi.org/10.5281/zenodo.14177191. DSI, direction selectivity index.

enhancement of direction selectivity is 70.15%±6.30% (**Fig 5G**), suggesting that an inter-lami-
nar direction-tuned gain mechanism plays a critical role in this process.

Subsequently, we analyzed the population-averaged direction tuning of the linear gain
extracted from Model II. We observed that the linear gain was direction-tuned, peaking in the
preferred direction (**Fig 6A**). A strong positive correlation was found between the DSI of the
linear gain and the DSI of the output layers (**Fig 6B**, $N = 17$ probe placements, $r = 0.57$,
$p = 0.0179$; Spearman's $\rho$), indicating that this direction-tuned gain mechanism contributes to
the enhancement of direction selectivity in the output layers. **Fig 6C** shows the comparison of
linear gains among the blank stimulus, null direction, and preferred direction. The gain in the
null direction exceeded that of the blank stimulus (**Fig 6C**, $N = 17$ probe placements,
$p = 0.00309$; Wilcoxon signed-rank test), suggesting interlaminar amplification of the response
in the null direction. Furthermore, the gain in the preferred direction was greater than those
for both the blank stimulus (**Fig 6C**, $N = 17$ probe placements, $p = 0.000503$; Wilcoxon signed-
rank test) and the null direction (**Fig 6C**, $N = 17$ probe placements, $p = 0.000293$; Wilcoxon
signed-rank test), demonstrating selective amplification in the preferred direction. Similar
results were obtained when considering only the signal transmission from L4 to L2/3 (**S3H**

**and S3I Fig**) and when using a different type of nonlinearity, the power-law nonlinearity (**S8 Fig**).

In the nonlinearity we used, whether a threshold-linear transfer function or a power-law nonlinearity, a threshold-like component is included, which could act like a suppression. To explicitly clarify the roles of linear gain and the suppression, we developed a third model. In this model (Model III), input layer responses are first multiplied by a linear gain, then subtracted by a constant, and finally processed through a rectified-linear nonlinearity (ReLU) to simulate output layer responses (**S9A Fig**). We found that Model II outperformed Model III by a very small margin (**S9B Fig**). In Model III, the gain was also direction-tuned (**S9D Fig**), and there was a strong relationship between the DSIs of the tuned gain and the output layers (**S9E Fig**).

Finally, we conducted GC analysis to identify the potential neural circuit origin of the direction-tuned gain. For a specific pair of connections, we calculated the pairwise-conditional GC value. In this calculation, all joint dependencies among the other known variables are conditioned out, effectively mitigating the impact of potential confounding factors, such as common inputs. Compared to blank stimuli, stimuli in the null direction elicited stronger feedforward connections from L4 to L2/3 and from L4 to L5 (**Fig 7A**). Moreover, stimuli in the preferred direction not only reinforced these connections from L4 to L2/3 and from L4 to L5 but also intensified the recurrent connections from L2/3 to L5 (**Fig 7B**). Notably, compared to those in the null direction, stimuli in the preferred direction further strengthened the connections

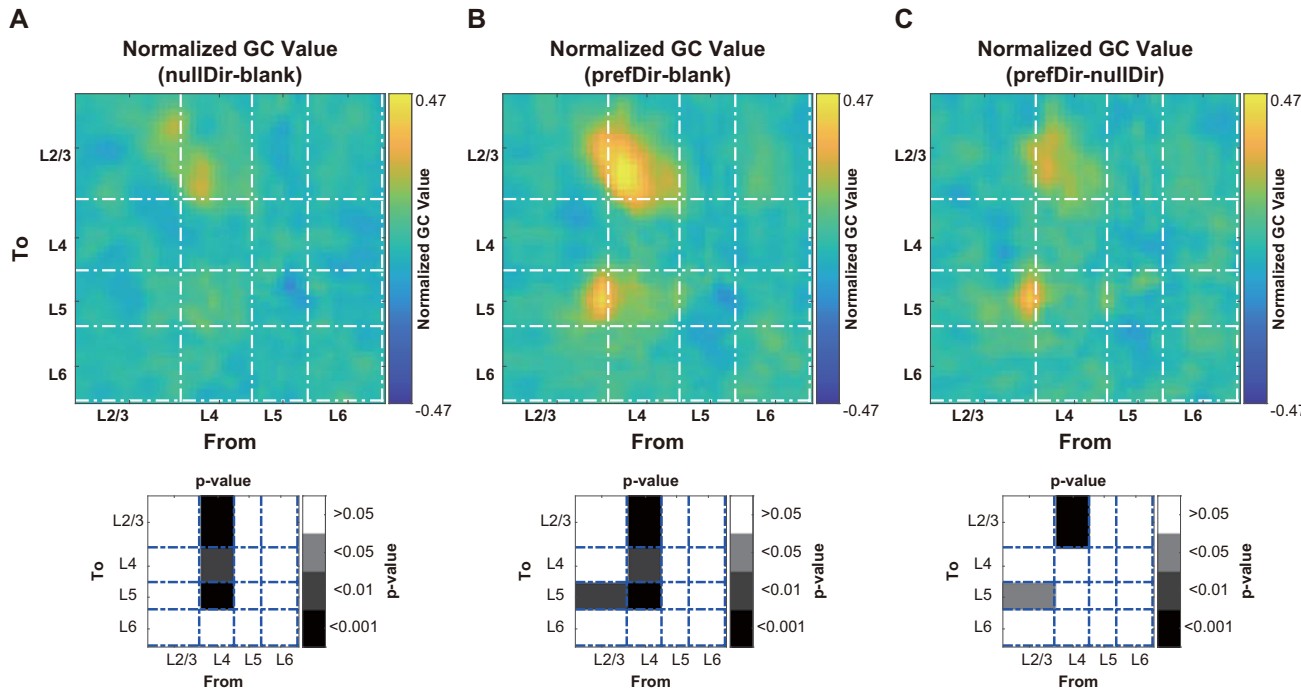

**Fig 7. The preferred direction evokes stronger feedforward connections from the input layers to the output layers and recurrent connections within the output layers. (A)** Differences in the normalized pairwise-conditional GC values between the null direction and blank stimulus. The pairwise-conditional GC value for a specific pair of connections is calculated by conditioning out all joint dependencies among the other known variables. This approach helps mitigate the impact of potential confounding factors, such as common inputs. The grayscale inset depicts the p value of the differences in GC values within each connection pair. The source layer (the sender of the projection, from) is positioned along the abscissa, while the target layer (the receiver of the projection, to) is represented along the ordinate. **(B)** Differences in the normalized GC values between the preferred direction and blank stimulus. **(C)** Differences in the normalized GC values between the preferred direction and the null direction. Data and code that support these findings are available at: https://doi.org/10.5281/zenodo.14177191. GC, Granger causality.

from L4 to L2/3 and from L2/3 to L5 (**Fig 7C**). **S10 Fig** shows the paired comparisons of normalized GC values between different stimulus conditions. These findings suggest that the direction-tuned gain likely stems from the combined effect of feedforward connections from the input layers to the output layers, as well as the recurrent connections within the output layers.

The gain of signal transmission from the input layers to the output layers exhibits direction tuning, which is likely to derive from the synergistic effects of both feedforward and recurrent connections. This direction-tuned gain, in addition to nonlinearity, enhances direction selectivity within the output layers, suggesting complex interplay between these neural pathways in processing motion information.

## Discussion

By leveraging laminar recording and computational modeling, we elucidated the processing and relay of direction selectivity across 2 successive stages of the visual hierarchy: the input layers and the output layers within cat early visual cortices. The strength of the direction selectivity is strongly correlated between the output layers and the input layers (**Fig 2C**) and preferred directions are preserved (**Fig 3**). Moreover, direction selectivity is enhanced in the output layers (**Fig 2**), where neuronal responses to the preferred direction are selectively preserved (**Fig 4**). The enhancement of direction selectivity likely results from direction-tuned signal transmission across layers (**Fig 6**), which may be attributed to both feedforward connections across input and output layers and recurrent connections within the output layers (**Fig 7**).

### Laminar processing of direction selectivity in the early visual cortices

In both carnivores [68,69] and primates [47], direction selectivity is considered an emergent property of the visual cortex and was initially observed in the V1 input layers. Unlike in cat early visual cortices, where direction-selective neurons are prevalent in all layers, in monkey V1 these neurons are only prevalent in the input layers (L4 and L6, [47]). This is because motion information is not primarily transmitted to the V1 output layers but is instead directed to the MT via L4B and L6 of V1 in monkeys [70]. Additionally, within the ventral pathway of monkey, motion information is transmitted to V2 thick stripes from L3 of V1 [71]. In contrast to monkey V1 [72,73], neurons in cat early visual cortices are organized into 2D direction maps [52–54,74–76]. These differences in the laminar and tangential organization of direction selectivity make the cat early visual cortices an ideal model for studying the laminar processing of direction selectivity.

Leveraging laminar recordings from multielectrode linear probes, our study characterized the laminar distribution of DSI and demonstrates, for the first time, that direction selectivity is enhanced in the output layers compared to the input layers within the same cortical column (**Fig 2**). Based on a database of 899 cells, Kim and colleagues characterized the distribution of DSI across different layers of cat area 17 (A17) [50]. They also found a higher proportion of direction-selective neurons in L2/3 and L5 compared to L4. In L6, however, they observed that the DSI distribution peaked at high DSI values, a pattern not observed in our data. Since our sample size of L6 neurons is relatively small compared to theirs, our sample may underrepresent some highly direction-selective neurons in L6. In a recent study, Hawken and colleagues identified 6 main functional clusters in L6 of monkey V1 [77], suggesting multiple parallel circuits within L6. One of these clusters, which is highly direction-selective, was located near the upper half of L6. Additionally, a two-photon study found that the direction map in cat A17 is not homogeneous but patchy [74]. These findings suggest that sampling at different cortical depths or positions may result in varying DSI distributions in L6.

Within a cortical column, we observed that a minority of the neuronal pairs demonstrated a shift in preferred directions of approximately 180 degrees (**Fig 3A**), corroborating earlier research [56]. The abrupt shift in preferred directions can be partially explained by the presence of direction fractures within the direction map [52–54,75] and the observation that some neurons exhibit opposite preferred directions for the 2 eyes [78] in cat early visual cortices. However, neuronal pairs with pronounced direction selectivity consistently maintain their preferred direction (**Fig 3B**). This suggests that strong direction selectivity is coupled with a more stable representation of motion information. Since the DSIs of the output layers and the interlaminar gain are positively correlated (**Fig 6B**), this stability may stem from the stronger direction-tuned gain of interlaminar signal transmission. These insights offer a vital point of reference for subsequent research on the plasticity of direction selectivity, as well as its potential modulation through learning or adaptive processes.

In addition to direction selectivity, other response properties also exhibited distinct laminar distributions, including the response modulation ratio (F1/F0) and orientation selectivity. Based on F1/F0, visual neurons are classified as simple cells (F1/F0 $\geq$ 1) and complex cells (F1/F0 < 1), with simple cells being more prevalent in the input layers and complex cells more abundant in the output layers (cat early visual cortices: [16]; monkey V1: [18]), as is shown in **S11 Fig**. Within both the simple and complex cell populations, direction selectivity is enhanced in the output layers (**S11C and S11D** Fig). Furthermore, we found no correlation between F1/F0 and DSI in either the output layers or the input layers (**S11E and S11F** Fig), in line with previous studies in cat early visual cortices [76] and monkey V1 [30,33]. These results suggest that the enhancement of direction selectivity is primarily due to interlaminar processing.

Similar to direction selectivity, previous studies have reported stronger orientation selectivity in the output layers compared to the input layers (cat early visual cortices: [79]; monkey V1: [17,18]), as demonstrated in **S12 Fig**. A strong correlation was found between orientation selectivity and direction selectivity, suggesting that the generation of orientation selectivity and direction selectivity may share partially overlapping mechanisms, such as intracortical inhibition [62,80,81] or spike threshold [19]. These diverse relationships between different response properties and their laminar dependence highlight the need for comprehensive large-scale models [82–84], which could provide insights into whether these properties share distinct mechanisms and the roles of diverse laminar circuits.

In this study, we combined data from cat area 17 (A17) and area 18 (A18). Previous studies have found a higher proportion of direction-selective neurons in A18 compared to A17 [85,86], with A18 displaying a homogeneous direction map [52,54] and A17 showing a patchy direction map [74]. Consistent with these findings, we also observed a higher proportion of direction-selective neurons in A18, but only in the output layers, not in the input layers (**S13 Fig**). This difference in DSI distributions within the output layers of A17 and A18 may be explained by the finding that, while the response ratios between the output and input layers are largest at the preferred direction in both areas, the response ratios in A18 tend to be higher than those in A17. It would be interesting to explore whether interlaminar processing varies between different cortical areas, such as V2 thick stripes [72,73] and MT [87] in monkey, where direction maps are also observed.

## Neural origin of direction-tuned gain

We identified 2 factors contributing to the enhanced direction selectivity in the output layers. The first is nonlinearity (**Fig 5**), potentially arising from gating of sensory information across layers [64] or nonlinear transformations from membrane potential to spike activity [63,88].

The second factor is a direction-tuned gain for interlaminar signal transmission (**Figs 5 and 6**). The functional relevance of this direction-tuned gain may lie in enhancing the signal-to-noise ratio in visual processing and allowing for more accurate and efficient encoding of motion information.

Through GC analysis, we demonstrated increased feedforward connections from the input layers to the output layers, as well as increased recurrent connections within the output layers at the preferred direction (**Fig 7**). The feedforward connections are inherently excitatory, while the increased recurrent connections could result from increased excitation, increased inhibition, or a combination of both.

The increase in excitation at the preferred direction may be supported by the presence of direction maps. Given the presence of direction maps, neurons within a direction column are likely to receive stronger excitatory inputs upon exposure to their preferred direction, which could result from enhanced feedforward convergence or recurrent connections, or both [89–91]. Future investigations are needed to determine whether the degree of enhancement in direction selectivity varies across different locations within the direction map and to explore whether the direction tuning of interlaminar signal transmission depends on these locations.

The involvement of inhibition in direction selectivity is evidenced by experiments showing the influence of direction selectivity through manipulations of inhibition or modeling results. Direction selectivity of cortical neurons is reported to decrease with the blockade of inhibition [92–96] and to increase with enhanced inhibition [80,97]. By modeling the cat's early visual cortices, Freeman revealed that inhomogeneities in the orientation map can cause spatiotemporal offsets between excitation and inhibition [98], and these offsets have been demonstrated to contribute to direction selectivity across various species (cat: [63]; monkey: [99]; ferret: [95]; mouse: [35,100]).

There are different possibilities for the relative direction tuning of inhibition and excitation: inhibition can be untuned [82], co-tuned with excitation [63,101–103], or tuned differently from excitation [95]. To study the role of inhibition in the interlaminar direction-tuned gain, future studies could use either pharmacology or optogenetics [95,104] to selectively activate or inactivate inhibition within certain cortical layers. If the impact of intracortical inhibition is significant, it would be beneficial to examine the roles of diverse types of inhibitory cells [105–107] and how the interplay between excitation and inhibition leads to the interlaminar direction-tuned gain.

In our single-unit data, when considering only narrow-spiking units, we found no significant difference in direction selectivity between the input and output layers, as direction selectivity was already strong in the input layers (**S4E Fig**). However, it is uncertain whether this finding can be attributed to the direction tuning of inhibitory neurons in cat early visual cortices, due to 2 factors: (1) the sample size of narrow-spiking units in the input layers is relatively small (28 single units); and (2) classification of neuronal types based on spike waveform remains controversial. Recent studies have shown that some narrow-spiking units in cat early visual cortices and monkey V1/premotor/motor cortex may belong to excitatory neurons [108–110], while some GABAergic neurons in mouse V1 exhibit broad-spiking waveforms [111]. Moreover, results on the direction selectivity of inhibitory neurons are diverse (cat early visual cortices [112,113]; ferret V1 [114]), highlighting the need for future studies to characterize the response properties of inhibitory neurons in the early visual cortices and their functional roles within neocortical networks.

In our model results, although responses in the output layers are derived by scaling responses from the input layers in our model (**Fig 5**), this does not conclude that the underlying neural circuit employs a pure multiplication algorithm. Alternatively, the tuned gain can be equivalent to a fixed gain combined with tuned suppression, as illustrated in Model IV of

S9 Fig. It has been shown that normalization—a process where an excitatory drive is divided by the summed activity from neighboring neurons—is a canonical neural computation [115–119] that can explain various neuronal response properties, including contrast gain control [120,121], cross-orientation suppression [122,123] and surround suppression [24,26,124]. Understanding whether normalization contributes to laminar processing of direction selectivity, and how interactions between excitatory and inhibitory neurons shape underlying neural circuits, is crucial for advancing our knowledge of visual information processing.

## Limitations of the current study

In the present study, we recorded neuronal responses in anesthetized animals, which is well-suited for studying cortical processing of basic visual features. However, the use of anesthetized animals presents 2 inherent limitations. First, anesthesia may attenuate feedback connections from higher visual areas [125–127]. These feedback connections may influence the direction selectivity of neurons in the early visual cortices [128,129]. Second, the impact of propofol, the anesthetic used in the present study, on neuronal responses throughout the cortical column is a matter of ongoing debate, and it remains unclear whether the differences in response strengths across layers are affected [130–133]. To address these limitations, future studies could record laminar responses in awake animals to determine whether the results are consistent with those of the present study or if additional connections are engaged in the absence of anesthesia.

By using drifting gratings with fixed stimulus properties such as contrast, size, spatial frequency, and temporal frequency, and by varying only the direction, we can directly compare direction selectivity between the input and output layers through simultaneous recordings. Previous studies have demonstrated that direction selectivity is modulated by varied spatiotemporal configurations of visual stimuli [7,134,135]. Additionally, both intralaminar and interlaminar connections are known to exhibit variability in response to changes in stimulus configurations [26,45,60,136,137]. Therefore, it would be worthwhile for future studies to employ a richer array of stimuli to determine whether the strategy for columnar processing of motion information varies across different spatiotemporal configurations.

As a caveat, while our results may primarily suggest a refinement of inherited direction selectivity in the output layers of the early visual cortices, we could not exclude the possibility of neurons exhibiting re-manufactured direction selectivity. Further research could elucidate whether such neurons exist, particular near the fractures of the direction map.

## STAR methods

**Animal preparation.** Acute experiments were performed in adult cats of both sexes (Felis catus, 2 to 4.5 kg; 9 animals). All procedures were performed in accordance with the National Institutes of Health Guidelines, and the research protocol was approved by the Biological Research Ethics Committee of Beijing Normal University (ID: IACUC(BNU)-NKLCNL 2017–06) and Fudan University (ID: 2021JS0087).

The details of the surgical procedure can be found in our previous studies [21,123,137]. In brief, animals were injected with dexamethasone (0.4 mg/kg, subcutaneously) 12 h before surgery. At the beginning of the surgery, the animals were initially anesthetized with isoflurane (5% concentration) and injected with atropine sulfate (0.05 mg/kg, subcutaneously). During the surgery and electrophysiological recording, anesthesia and paralysis were maintained with propofol (2 to 6 mg/kg/h) and vecuronium bromide (0.1 mg/kg/h), respectively. The end-tidal $CO_2$ concentration was maintained at 3.5% to 4%, and the body temperature was maintained at 36.5˚C to 37.5˚C. Ventilation pressure, heart rate, electrocardiogram, and blood oxygen

were monitored continuously. To prevent brain edema and to reduce saliva secretions during the experiment, dexamethasone was injected every day (0.4 mg/kg), and atropine sulfate (0.05 mg/kg) was injected every other day. To dilate the pupils, 1% atropine sulfate solution was administered. To protect the corneas and focus visual stimuli on the retina, contact lenses with appropriate refractive power (+2.0 D) were fitted onto the eyes. To reduce optical aberration, artificial pupils (3 mm in diameter) were placed in front of the eyes. Before the experiment, the optic disc was back-projected onto the screen with a reversible ophthalmoscope, from which we estimated the location of the area centralis.

The animal's head was fixed on the stereotaxic instrument, and a craniotomy was performed to enable electrophysiological recording of the early visual cortices (P7-A5, L0-L5; Horsley–Clarke coordinates, both hemispheres, including A17 and A18). After the penetration of the linear probe, 1.5% to 2% agar was applied to protect the cortex.

### Electrophysiological recording

A multielectrode linear array was used to record neuronal activity across all layers simultaneously. A 24-electrode linear probe (U-Probe, Plexon; 15 μm in diameter and 100 μm interelectrode distance) was used for all animals except for one animal, in which a 64-electrode linear probe (Probe64D Sharp, IDAX Microelectronics; 50 μm interelectrode distance) was used. The linear probe was controlled by a microelectrode drive (NAN Instruments, Israel) to penetrate across the cortex. Electrical signals were acquired at a sampling rate of 30,000 Hz with a Cerebus 128-channel system (Blackrock Microsystems).

To obtain the multiunit spiking activity (MUA), we first high-pass filtered the raw data (seventh order Butterworth with a 1,000 Hz corner frequency). Subsequently, we thresholded the filtered data based on a signal-to-noise ratio (SNR) of 5.5, setting responses that exceeded this threshold to 1, while all others were set to 0. The raw data were also low-pass filtered (seventh order Butterworth with a 300 Hz corner frequency) to obtain the LFP. Both the MUA and LFP were further downsampled to 500 Hz.

In total, we obtained 26 probe placements that had a high SNR and were perpendicular to the cortex. The perpendicularity of each probe placement was verified based on the variation of RF center positions and the consistency of preferred orientations across recording sites. To quantify the variation of RF center positions, we calculated the RF Center Dispersion:

$$\text{RF Center Dispersion} = \frac{\frac{1}{N}\sum_{i=1}^{N}\left((x_i - \bar{x})^2 + (y_i - \bar{y})^2\right)}{2 * \sqrt[N]{\prod_{i=1}^{N}\sigma_i}}, \tag{1}$$

where $(x_i, y_i)$ and $\sigma_i$ are the center position and radius of the $i$th recording site, $N$ is the number of recording sites in one probe placement, and $(\bar{x}, \bar{y})$ is the averaged center position across $N$ probe placements. This parameter quantifies the variation in RF centers relative to the RF size within one probe placement, with the numerator representing the variation of RF center positions and the denominator representing the population-averaged RF diameter. A smaller value indicates less variation in RF center positions across all recording sites.

We quantified the consistency of preferred orientations by calculating Preferred Orientation Consistency:

$$\text{Preferred Orientation Consistency} = \frac{1}{N}\left|\sum_{i=1}^{N} e^{i*2*\theta_i}\right|, \tag{2}$$

where $\theta_i$ is the preferred orientation of the $i$th recording site and $|C|$ is the amplitude of a complex value. The range for Preferred Orientation Consistency is from 0 to 1, with higher values

indicating greater consistency in preferred orientations across recording sites; a value of 1 occurs when all recording sites within a probe placement share the same preferred orientation. In **S1A Fig**, we presented RF Center Dispersion and Preferred Orientation Consistency for all probe placements. Low RF Center Dispersion values and Preferred Orientation Consistency values close to 1 confirm perpendicular penetration of these probes. For long-lasting probe placements, perpendicularity was further verified postmortem through histology, as probe tracks were clearly delineated by CO staining.

## Visual stimulation

Visual stimuli were generated on a PC with a Leadtek GeForce 6800 video card and were presented on a gamma-calibrated CRT monitor (Dell P1230, refresh rate 100 Hz, mean luminance 32 cd/$m^2$). The stimuli were all monocularly presented to the eye contralateral to the recorded hemisphere.

In the present study, 2 types of stimuli were employed: sparse noise stimuli (SPN) and drifting gratings. Before utilizing the drifting gratings to determine the direction selectivity, the SPN was used to identify the center and the size of the classical RF of each recording site. Furthermore, the layers were assigned based on neuronal responses to the SPN.

The sparse noise stimuli consisted of black and white squares presented at random positions on a 2D grid. Each square had a 90% contrast from the uniform gray background (luminance: 32 cd/$m^2$) and was presented for 40 ms in a reverse-correlation paradigm. The size of each square, the distance between neighboring squares, and the extent of the visual space covered were chosen based on the classical RF size of the recording sites (typically 1–5 degrees).

Drifting gratings, similar to the SPN, had a 90% contrast and a mean luminance of 32 cd/$m^2$. The size of each grating was slightly larger than the classical RF mapped using the SPN. The temporal frequency was set at 2 Hz, with the duration ranging from 2 to 4 s. We chose spatial frequencies that elicit strong responses for each probe placement, ranging from 0.05 cpd to 0.4 cpd. During the recordings for each probe placement, only the direction of the gratings was varied, while all the other parameters remained constant. The number of directions tested for each probe placement varied from 12 to 24. In each direction, the images were presented 5 to 100 times, and 5 to 10 repetitions were typically sufficient to achieve a high SNR. Among the 26 probe placements, 12, 18, and 24 directions were used 6, 15, and 5 times, respectively, while 5, 10, and 100 repetitions were used 1, 20, and 5 times, respectively. Experiments involving 100 repetitions were designed to investigate additional research questions. In addition to grating stimuli, blank stimuli (a uniform gray background) will randomly appear in some trials. These trials with blank stimuli constituted approximately 10% of the total trials.

## Laminar assignment

For each probe placement, we assigned all recording sites to corresponding layers according to stimulus-evoked MUAs, LFPs, CSDs, and spatial RFs from SPN experiments [21].

Concerning the differences in cortex thickness across probe placements, we assigned a relative depth to each recording site [41,42,138,139]. The range of relative depth was from 0 to 1, with 0 indicating the border between the cortical surface and L2/3, and 1 indicating the border between L6 and white matter. The border between the cortical surface and L2/3 can always be identified from LFPs because the LFP waveform is nearly identical for recording sites out of the cortex, and an abrupt change exists between neighboring recording sites once it enters the cortex. The border between L6 and white matter was determined by MUA and spatial RF. The deepest site with a high SNR of MUA and consecutive spatial RF was deemed 50 μm above this border.

We separated the cortex into 4 layers (L2/3, L4, L5, and L6) using 3 relative depths (0.35, 0.58, and 0.76 from top to bottom) to mark the borders between neighboring layers. These borders were determined primarily based on MUA response latency, defined as the time point at which the response reached half of its maximum (**Fig 1B**). Notably, L4 and L6 exhibited faster latencies than L2/3 and L5 (**Fig 1C–1E**).

In **S2A Fig**, we showed MUA, LFP, and CSD responses evoked by black and white stimuli within the spatial RF for an example probe placement. MUA responses revealed advances in latency and a stronger rebound in both L4 and L6 [21]. LFP responses indicated a polarity reversal in L6, while CSD responses identified a current sink in L4 [140,141]. **S2B Fig** displays the number of electrodes within each layer across all probe placements, consistent with the relative thickness of each layer.

To enhance the statistical power, we combined the data from L2/3 and L5 as the "output layers" and those from L4 and L6 as the "input layers" due to their roles in signal transmission along the visual hierarchy [41] and their similarities in response properties [21]. In **S3A Fig**, we further showed that the DSI in L2/3 and L5 was significantly larger than in L4 and L6. Similar results were observed when comparing between L4 and L2/3, as seen between the output layers and input layers, including enhanced direction selectivity in L2/3 (**S3B and S3C Fig**), a direction-tuned response ratio between L2/3 and L4 (**S3D–S3G Fig**), and a direction-tuned gain mechanism from L4 to L2/3 (**S3H and S3I Fig**).

## Data analysis

**A. Selection of recording sites.**   For each recording site, we calculated the trial-averaged MUA response to each test stimulus direction, represented by $\theta$, and denoted it as $R(\theta, t)$. For each stimulus direction, $R(\theta, t)$ can be plotted as a peristimulus time histogram (PSTH). Similarly, we obtained $R(Blank, t)$ for blank stimuli. We defined $Var(\theta)$ and $Var(Blank)$ as the variances of $R(\theta, T)$ and $R(Blank, T)$, respectively, where $T$ includes the 0 to 200 ms period after stimulus onset. From these measurements, we calculated the SNR of each recording site as follows:

$$SNR = \frac{\max(Var(\theta))}{Var(Blank)}.$$

(3)

Only recording sites with an SNR larger than 8 were further analyzed.

**B. Quantification of direction selectivity and orientation selectivity.**   We fitted direction tuning curves of the time-averaged response, $R(\theta)$, with the von Mises function [142].

$$R(\theta) = \alpha \frac{e^{(\sigma_1 \cdot cos(\theta - \theta_{pref}))} - e^{-\sigma_1}}{e^{\sigma_1} - e^{-\sigma_1}} + \beta \frac{e^{(\sigma_2 \cdot cos(\theta - \theta_{null}))} - e^{-\sigma_2}}{e^{\sigma_2} - e^{-\sigma_2}} + R_0.$$

(4)

In the above equation, $\theta_{pref}$ and $\theta_{null}$ represent the preferred direction and null direction, respectively, and they are always of 180° difference. $\sigma_1$ and $\sigma_2$ control the tuning width in the preferred direction and null direction, respectively. For initial-guess values, we set $\theta_{pref}$ as the direction eliciting the maximum response, $\sigma_1$ and $\sigma_2$ to 1, $\alpha$ and $\beta$ as the difference between the maximum and minimum responses across all directions, and $R_0$ as the minimum response across all directions. The optimal parameters are then obtained using fmincon in MATLAB.

Based on neuronal responses in the preferred direction $R(\theta_{pref})$ and in the null direction $R(\theta_{null})$, we calculated the DSI as follows:

$$DSI = \frac{R(\theta_{pref}) - R(\theta_{null})}{R(\theta_{pref}) + R(\theta_{null})}. \tag{5}$$

The DSI ranges from 0 to 1, with a larger value indicating stronger direction selectivity.
In addition to direction tuning curves, we also fitted orientation tuning curves as:

$$R(\theta) = \alpha \frac{e^{(\sigma_1 \cdot \cos(2*(\theta - \theta_{pref})))} - e^{-\sigma_1}}{e^{\sigma_1} - e^{-\sigma_1}} + R_0. \tag{6}$$

In the above equation, $\theta_{pref}$ represents the preferred orientation. We defined $\theta_{ortho}$ as the orthogonal orientation, which is always 90° away from $\theta_{pref}$.

Based on the fitted orientation tuning, we get responses at the preferred orientation $R(\theta_{pref})$ and at the orthogonal orientation $R(\theta_{ortho})$. We defined the orientation selectivity index (OSI) as follows:

$$OSI = \frac{R(\theta_{pref}) - R(\theta_{ortho})}{R(\theta_{pref}) + R(\theta_{ortho})}. \tag{7}$$

We further calculated OP ratio as the ratio between $R(\theta_{ortho})$ and $R(\theta_{pref})$ and HWHH as the half-width at half-height.

**C. Model fitting.** By simulating the signal transmission from the input layers to the output layers, we constructed 2 computational models to fit the neuronal responses in the output layers. We assumed that there are 2 stages for interlaminar signal transmission. The first stage is a linear gain, and the second stage is a static nonlinearity.

In the first model (Model I, untuned gain), the linear gain was constant under all stimulus conditions, including different directions and blank stimuli.

$$\hat{R}_{(s)}^{output\ layer,i} = f(w R_{(S)}^{input\ layer,i}). \tag{8}$$

$R_{(S)}^{input\ layer,i}$ is the average neuronal response across all the valid recording sites within the input layers at the $i$th repetition. We use the subscript $S$ to denote all stimuli used, which include both grating stimuli and blank stimuli. $\hat{R}_{(s)}^{output\ layer,i}$ is the output response. $w$ is the fixed linear gain across all stimulus conditions, and $f$ is a static threshold-linear nonlinearity with the following form:

$$f(g) = \left(\frac{A(g - g_0)}{2}\right)\left(1 + erf\left(\frac{g - g_0}{\sqrt{2}\sigma}\right)\right) + \frac{A\sigma}{\sqrt{2\pi}} e^{\left(-\frac{(g - g_0)^2}{2\sigma^2}\right)}. \tag{9}$$

In the above equation, $g_0$ is the threshold, $A$ is the gain, and $\sigma$ controls the smoothness of the transition zone, with larger values generating smoother transitions. The derivation and examples of this equation can be found in previous studies [65–67].

In the second model (Model II, tuned gain), the linear gain, $w_{(S)}$, varied under different stimulus conditions.

$$\hat{R}_{(s)}^{output\ layer,i} = f(w_{(S)} R_{(S)}^{input\ layer,i}). \tag{10}$$

In both models, we optimized parameters to minimize the residual between the output

responses $\hat{R}_{(s)}^{output\ layer,i}$ and the true neuronal responses in the output layers $R_{(s)}^{output\ layer,i}$.

$$J = \sum_S \sum_i (\hat{R}_{(s)}^{output\ layer,i} - R_{(s)}^{output\ layer,i})^2. \tag{11}$$

To compare Model I and Modell II, we used the adjusted goodness of fit ($adjR^2$), which accounts for the number of free parameters in each model:

$$adjR^2 = 1 - \frac{N_{sample} - 1}{N_{sample} - N_{param}} * \frac{2*(\hat{R} - R)^2}{(\hat{R} - \bar{\hat{R}})^2 + (R - \bar{R})^2}. \tag{12}$$

In the above equation, $N_{sample}$ and $N_{param}$ represent the number of data samples and free parameters, respectively. $\hat{R}$ and $R$ are the fitted and raw response, respectively. $\bar{\hat{R}}$ and $\bar{R}$ are the average value of the fitted and raw responses, respectively.

In addition to the form of nonlinearity in Eq (9), we also used a power-law nonlinearity:

$$f(g) = A[g - g_0]_+^n. \tag{13}$$

In the above equation, $g_0$ is the threshold, $A$ is the gain, and $[C]_+$ represents half rectification.

In the above 2 forms of nonlinearity, there is a suppressive component in each. To explicitly study the role of linear gain and suppression, we developed 2 additional models (Model III and Model IV). These models incorporate 3 stages: a linear gain, a suppressive component, and a rectified-linear nonlinearity (ReLU):

$$\hat{R}_{(s)}^{output\ layer,i} = [w_{(S)} R_{(S)}^{input\ layer,i} - T_{(S)}]_+. \tag{14}$$

In the above equation, $T_{(S)}$ represents the suppressive component. In Model III, $w_{(S)}$ varied at each stimulus condition, while $T_{(S)}$ was constant. Conversely, in Model IV, $w_{(S)}$ remained constant and $T_{(S)}$ varied at each stimulus condition.

**D. Granger causality analysis.** Multivariate Granger causality was calculated using the MVGC MATLAB Toolbox [143], and the details can be found in our previous work [21]. For each probe placement, we obtained the pairwise-conditional GC value, $GC(S, E_{to}, E_{from})$, under all stimulus conditions $S$, where $E_{to}$ and $E_{from}$ were the targets and sources of the connections, respectively. When calculating the pairwise-conditional GC value for a specific pair of connections, all joint dependencies among the other known variables are conditioned out. This approach helps mitigate the impact of potential confounding factors, such as common inputs. We then z-scored $GC(S, E_{to}, E_{from})$ and obtained the normalized GC value.

## Spike sorting

Spike sorting was initially performed automatically using KiloSort [144] and then manually curated using Phy [145]. The quantity of each unit was quantified by the SNR of the spike waveform, which was defined as the ratio of the peak-to-peak amplitude of the mean waveform to twice the standard deviation of the noise [146]. Any unit with a spike waveform SNR greater than 1.5 was classified as a single unit (SU).

We separated all SUs into 5 classes (narrow-spiking units, broad-spiking units, triphasic-spiking units, compound-spiking units, and positive-spiking units) based on recent work studying extracellular spike waveforms in cat early visual cortices [147]. The details of the classification can be found in our previous work [21].

For each SU, we calculated the response modulation ratio (F1/F0) under the preferred stimulus condition, where F1 represents the response at the same temporal frequency as the stimulus drifting, and F0 is the averaged response. We then categorized each SU as either a simple

cell (F1/F0 $\geq$ 1) or a complex cell (F1/F0 < 1). The laminar profile of F1/F0 and its relationship with DSI are illustrated in **S11 Fig**.

## Experimental design and statistical analysis

The median values of all recording sites within the input and output layers from one probe placement were used for interlaminar comparison, and under these conditions, an error-bar indicating the SEM is also shown. Otherwise, each recording site was used as an independent repetition. To test whether there was a difference between the medians of 2 groups, we used the Wilcoxon signed-rank test for paired groups and the Wilcoxon rank-sum test for independent groups. Unless explicitly stated, a two-sided test was used. The Kruskal–Wallis test, employed to compare medians across multiple groups, was supplemented with the Bonferroni method to adjust for multiple comparisons. To determine if the median of a group was different from 0, we used a sign test. The chi-square goodness-of-fit test was used to compare the proportions of 2 groups. The Spearman correlation coefficient was used to test the correlation between 2 variables. All error bars represent the median ± SEM (standard error of the mean), unless otherwise noted.

## Supporting information

**S1 Fig. Verification of perpendicular penetration for each probe placement. (A)** Dispersion of RF centers and consistency of preferred orientation of all probe placements. RF Center Dispersion (the abscissa) is calculated as $\frac{\frac{1}{N}\sum_{i=1}^{N}((x_i-\bar{x})^2+(y_i-\bar{y})^2)}{2*\sqrt[N]{\prod_{i=1}^{N}\sigma_i}}$, where $(x_i, y_i)$ and $\sigma_i$ are the center position and radius of the $i$th recording site, $N$ is the number of recording sites in one probe placement, and $(\bar{x}, \bar{y})$ is the averaged center position across $N$ probe placements. The numerator represents the variance of RF center positions, and the denominator is the population-averaged RF diameter. A smaller value indicates less variation in RF centers across all recording sites. Preferred Orientation Consistency (the ordinate) is calculated as $\frac{1}{N}\left|\sum_{i=1}^{N}e^{i*2*\theta_i}\right|$, where $\theta_i$ is the preferred orientation of the $i$th recording site and $|C|$ is the amplitude of a complex value. A larger value indicates higher consistency in preferred orientations across all recording sites. Each dot represents 1 probe placement. Results from 3 example probe placements are shown in **B–D**. **(B–D)** Example probe placements with RF Center Dispersion and Preferred Orientation Consistency as indicated in **A**. Red and blue lines represent recording sites in the input layers and the output layers, respectively. Data and code that support these findings are available at: https://doi.org/10.5281/zenodo.14177191.
(EPS)

**S2 Fig. Layer assignment. (A)** Temporal responses of MUA, LFP, and CSD to black and white stimuli within the RF center from an example probe placement. MUA and LFP responses were normalized by their maximum absolute response over time at each recording site. MUA responses show that the input layers (L4 and L6) leads the output layers (L2/3 and L5) and exhibits a stronger rebound [21]. LFP and CSD responses indicate a latency change between L2/3 and L4 and a polarity reversal between L5 and L6. **(B)** Distribution of electrode number within each layer from all probe placements. For the U-Probe (100-μm interelectrode distance), electrode numbers in the output layers (L2/3 and L5) are shown in blue, and those in the input layers (L4 and L6) are shown in red. For Probe64D Sharp (50-μm interelectrode distance), electrode numbers are shown in black. Data and code that support these findings are available at: https://doi.org/10.5281/zenodo.14177191.
(EPS)

**S3 Fig. Enhanced direction selectivity in L2/3 and an interlaminar direction-tuned gain from L4 to L2/3. (A)** Comparison of DSIs across all layers. The DSIs in L2/3 and L5 were significantly greater than those in L4 and L6. **(B)** Cumulative distributions of DSIs in L2/3 (blue line) and L4 (red line). The horizontal dashed line indicates a cumulative proportion of 0.5, and the vertical dashed line represents a DSI of 1/3. The median value and SEM of the DSI for both layers are shown at the top. The proportions of recording sites exhibiting strong direction selectivity (DSI > 1/3) with the input and output layers are denoted by colored arrows on the right. Statistical comparisons for median DSI values (at the top) and proportions of recording sites with strong direction selectivity (on the right) between the output layers and the input layers are also shown. **(C)** Comparison of DSIs between L2/3 and L4 within the same probe placement. For each probe placement, we obtained the median value and SEM of the DSI across all recording sites in L2/3 and in L4. Thus, each dot represents data from 1 probe placement. The Spearman correlation and the significance of paired comparisons between L2/3 and L4 are shown. Data are median ± SEMs. **(D)** Population-averaged direction tuning of response strength within L2/3 (blue lines) and L4 (red lines). The blue and red horizontal lines represent the average responses to blank stimuli within L2/3 and L4, respectively. Data are mean ± SEMs. **(E)** Population-averaged direction tuning of the response ratio between L2/3 and L4. Data are mean ± SEMs. **(F)** Distribution of the response ratios under the indicated stimulus conditions. The vertical dashed line represents a response ratio value of 1. From left to right, the panels show the response ratio under blank stimulus, in the null direction, and in the preferred direction. **(G)** Comparison of the response ratios between the indicated stimulus condition pairs. From left to right, the panels depict the response ratio between the null direction and blank stimulus, between the preferred direction and blank stimulus, and between the preferred direction and the null direction. Each dot indicates the data from one probe placement. **(H)** Population-averaged direction tuning of linear gains in Model II when considering only the signal transmission from L4 to L2/3. The horizontal line represents the linear gain under the blank stimulus. Data are mean ± SEMs. **(I)** Comparison of linear gains between the indicated stimulus condition pairs. From left to right, the panels depict the comparison of linear gains between the null direction and the blank stimulus, between the preferred direction and the blank stimulus, and between the preferred direction and the null direction. Data and code that support these findings are available at: https://doi.org/10.5281/zenodo.14177191. (EPS)

**S4 Fig. Enhanced direction selectivity in the output layers and maintenance of preferred directions across layers for sorted single units. (A)** Spike waveforms and proportions of each type of sorted single unit (SU). NS, narrow-spiking units; BS, broad-spiking units; TS, triphasic-spiking units; CS, compound-spiking units; PS, positive-spiking units. **(B)** Comparison of DSIs between the output layers and the input layers within the same probe placement from all sorted single units. Within the same cortical column, direction selectivity was enhanced in the output layers ($N$ = 24 probe placements, $p$ = 0.0036; Wilcoxon signed-rank test), and the DSIs between the output layers and the input layers were highly correlated ($r$ = 0.4783, $p$ = 0.0192; Spearman's $\rho$). Each dot represents data from one probe placement. Data are median ± SEMs. **(C)** Results from all sorted single units. Left: Cumulative distributions of DSIs in the output layers (blue line) and the input layers (red line). Compared to the input layers, the output layers exhibited a significantly larger DSI (output layers: 0.5612+0.0218, $N$ = 145 single units; input layers: 0.3663+0.0251, $N$ = 118 single units; interlaminar difference: $p$ = 0.0011; Wilcoxon rank-sum test) and had a larger proportion of recording sites with strong direction selectivity (proportion of recording sites with DSI > 1/3, output layers: 73.79%, $N$ = 145 single

units; input layers: 55.08%, *N* = 118 single units; interlaminar difference: *p* = 0.0015; chi-square goodness-of-fit test). Right, differences in preferred directions between single units in the output layers and the input layers. **(D)** Similar to C, but for single units excluding narrow-spiking units. Compared to the input layers, the output layers exhibited a significantly larger DSI (output layers: 0.5144+0.0271, *N* = 95 single units; input layers: 0.3250+0.0274, *N* = 90 single units; interlaminar difference: *p* = 0.0136; Wilcoxon rank-sum test) and had a larger proportion of recording sites with strong direction selectivity (proportion of recording sites with DSI > 1/3, output layers: 70.53%, *N* = 95 single units; input layers: 48.89%, *N* =90 single units; interlaminar difference: *p* = 0.0027; chi-square goodness-of-fit test). **(E)** Similar to C, but for narrow-spiking units only. The DSIs of the output and input layers showed no significant difference (output layers: 0.6582+0.0348, *N* =50 single units; input layers: 0.5083+0.0555, *N* =28 single units; interlaminar difference: *p* = 0.1708; Wilcoxon rank-sum test), with a similar proportion of recording sites exhibiting strong direction selectivity (proportion of recording sites with DSI > 1/3, output layers: 80%, *N* = 50 single units; input layers: 75%, *N* = 28 single units; interlaminar difference: *p* = 6079; chi-square goodness-of-fit test). Data and code that support these findings are available at: https://doi.org/10.5281/zenodo.14177191.
(EPS)

**S5 Fig. Interlaminar comparison of response strength to blank stimuli, in the null direction and in the preferred direction. (A)** Cumulative distribution of response strength to blank stimuli in the output layers (blue line) and the input layers (red line). The median response strength to blank stimuli is stronger in the input layers than in the output layers. **(B)** Comparison of response strength to blank stimuli between the output layers and the input layers within the same probe placement. The response strength to blank stimuli is weakened from the input layers to the output layers within the same cortical column. Each dot represents the data from one probe placement. Data are median ± SEMs. **(C, D)** Similar to **A** and **B** but for response strength in the null direction. The response strength to the null direction is weakened in the output layers. **(E, F)** Similar to **A** and **B** but for response strength in the preferred direction. The response strength to the preferred direction shows no significant difference between the output layers and the input layers. Data and code that support these findings are available at: https://doi.org/10.5281/zenodo.14177191.
(EPS)

**S6 Fig. Model selection based on the Akaike information criteria (AIC) and Bayesian information criteria (BIC). (A)** Comparison of models based on AIC. Compared to Model I (Untuned Gain, abscissa), the AIC of Model II (Tuned Gain, ordinate) is smaller, indicating that Modell II can explain the data better. **(B)** Similar to **A**, but based on BIC. Model II (ordinate) also has smaller BIC than Model I (abscissa), indicating that a model with direction-tuned gain can explain the data better. Data and code that support these findings are available at: https://doi.org/10.5281/zenodo.14177191.
(EPS)

**S7 Fig. Model II accurately replicates the relationships of response ratios between different stimulus conditions as observed in the data. (A)** Relationship of response ratios between different stimulus conditions in the data: no correlation between the null direction and the blank condition (left), a negative correlation between the preferred direction and the blank condition (middle), and a positive correlation between the preferred and null directions (right). **(B)** Similar to **A**, but for results from Model II, which accurately replicates the relationships of response ratios between different stimulus conditions as observed in the data. Data and code that

support these findings are available at: https://doi.org/10.5281/zenodo.14177191.
(EPS)

**S8 Fig. Similar results are observed with a power-law nonlinearity as with a threshold-linear nonlinearity for the models. (A)** Comparison of DSIs between the output of Model I and the input layers. In this figure, a power-law nonlinearity is used in Model I and Model II instead of a threshold-linear nonlinearity. **(B)** Comparison of DSIs between the output of Model I and the output layers. **(C)** Comparison of adjusted goodness of fit ($adjR^2$) between Model I and Model II. The horizontal and vertical dashed lines represent an $adjR^2$ value of 0.8. **(D)** Comparison of DSIs between the output of Model II and the output layers. **(E)** Population-averaged direction tuning of linear gains in Model II. The horizontal line is the linear gain under the blank stimulus. Data are mean ± SEMs. **(F)** Relationship of DSIs between the linear gain and the output layers. Data and code that support these findings are available at: https://doi.org/10.5281/zenodo.14177191.
(EPS)

**S9 Fig. Similar performance as Model II for models with a gain-suppression cascade. (A)** Model structure. Left, Model II in the main text. Middle and right, neuronal responses in the input layers are first multiplied by a linear gain ($G$), then subtracted by a suppressive component ($S$), and finally passed through a rectified-linear nonlinearity (ReLU) to simulate output layer responses. In Model III (middle panel), the linear gain is direction-tuned while the suppression is untuned to direction. In Model IV (right panel), the linear gain is untuned to direction and the suppression is direction-tuned. The exemplar neuronal responses in the input and output layers are the same as those in **Fig 4A**, and are also presented as mean ± SEMs. **(B)** Comparison of adjusted fitting goodness ($adjR^2$) across Model II and Model III. Model II outperforms Model III by a very small margin ($adjR^2$ of Model II VS $adjR^2$ of Model III: 0.9502 ±0.0101 VS 0.9429±0.0096). **(C)** Comparison of adjusted fitting goodness ($adjR^2$) across Model II and Model IV. Model II and Model IV shows no significant difference ($adjR^2$ of Model II VS $adjR^2$ of Model IV: 0.9502±0.0101 VS 0.9492±0.0073). **(D)** Population-averaged direction tuning of the gain in Model III. **(E)** The DSI of gain in Model III highly correlates with that of the output layers. Data and code that support these findings are available at: https://doi.org/10.5281/zenodo.14177191.
(EPS)

**S10 Fig. Comparison of normalized GC values between different stimulus conditions. (A)** Comparison of normalized GC values between null direction and blank condition. From left to right, GC values are calculated as follows: from L4 to L2/3, from L4 to L4, from L4 to L5, and from L2/3 to L5, respectively. The significance level has been adjusted using the Bonferroni method to account for multiple comparisons, with the *p*-value shown being multiplied by the number of comparisons (4 * 4 = 16 comparisons). **(B)** Comparison of normalized GC values between preferred direction and blank condition. **(C)** Comparison of normalized GC values between preferred direction and null direction. Data and code that support these findings are available at: https://doi.org/10.5281/zenodo.14177191.
(EPS)

**S11 Fig. Direction selectivity strength shows no correlation with the modulation ratio. (A)** Laminar pattern of the F1/F0. The vertical dashed line represents a F1/F0 of 1, indicating that the averaged response (F0) and the modulated response at the stimuli's temporal frequency (F1) are of the same strength. **(B)** Cumulative distributions of F1/F0 in the output layers (blue line) and the input layers (red line). The horizontal dashed line indicates a cumulative proportion of 0.5, and the vertical dashed line represents a F1/F0 of 1. The modulation ratio is

significantly larger in the input layers than in the output layers. **(C)** Cumulative distributions of DSI for all simple cells (F1/F0 $\geq$ 1) in the output layers (blue line) and the input layers (red line). Direction selectivity is significantly increased from the input layers to the output layers for simple cells. **(D)** Similar to C but for results from complex cells (F1/F0 < 1). **(E)** DSI shows no significant correlation with F1/F0 in the output layers. **(F)** DSI shows no significant correlation with F1/F0 in the input layers. **(G)** Cumulative distributions of DSI for all simple cells (solid line) and complex cells (dashed line) in the output layers. There exists no significant difference in direction selectivity strength between simple cells and complex cells within the output layers. **(H)** Similar to G but for results from the input layers. Data and code that support these findings are available at: https://doi.org/10.5281/zenodo.14177191.
(EPS)

**S12 Fig. Strong correlation between direction selectivity and orientation selectivity. (A)** Laminar pattern of the orientation selectivity index (OSI). The OSI was calculated as $\frac{R(prefOri)-R(orthoOri)}{R(prefOri)+R(orthoOri)}$, where $R(prefOri)$ and $R(orthoOri)$ are the neuronal responses in the preferred orientation and in the orthogonal orientation, respectively. **(B)** Cumulative distributions of OSI in the output layers (blue line) and the input layers (red line). The orientation selectivity is significantly stronger in the output layers than in the input layers. **(C)** DSI are strongly correlated with OSI in both the output layers (left) and the input layers (right). **(D–F)** Similar as **A–C**, but for the results of OP ratio. The OP ratio was calculated as $\frac{R(orthoOri)}{R(prefOri)}$. **(G–I)** Similar as **A–C**, but for the results of HWHH. HWHH is defined as the half-width at half-height of the maximum response. Data and code that support these findings are available at: https://doi.org/10.5281/zenodo.14177191.
(EPS)

**S13 Fig. Laminar processing of direction selectivity is consistent across areas 17 and 18. (A–C)** Results from area 17 (A17); **(D–F)** results from area 18 (A18); **(G–I)** results combining data from A17 and A18. **(A, D, and G)** Cumulative distributions of DSIs in the output layers (blue lines) and the input layers (red lines). Compared to the input layers, the output layers exhibited a significantly larger DSI (A17: output layers: 0.4035+0.0213, $N$ = 67 single units; input layers: 0.2464+0.0263, $N$ = 45 single units; interlaminar difference: $p$ = 5.8990*10$^{-4}$; A18: output layers: 0.5228+0.0228, $N$ = 67 single units; input layers: 0.2598+0.0274, $N$ = 64 single units; interlaminar difference: $p$ = 2.1344*10$^{-6}$; A17 and A18: output layers: 0.4605±0.0161, $N$ = 134 recording sites; input layers: 0.2566±0.0194, $N$ = 109 recording sites; interlaminar difference: $p$ = 1.7392*10$^{-8}$; Wilcoxon rank-sum test) and had a larger proportion of recording sites with strong direction selectivity (proportion of recording sites with DSI > 1/3, A17: output layers: 68.66%, $N$ = 67 single units; input layers: 42.22%, $N$ = 45 single units; interlaminar difference: $p$ = 0.0054; A18: output layers: 77.61%, $N$ = 67 single units; input layers: 43.75%, $N$ = 64 single units; interlaminar difference: $p$ = 7.0902*10$^{-5}$; A17 and A18: output layers: 73.13%, $N$ = 134 recording sites; input layers: 43.12%, $N$ = 109 recording sites; interlaminar difference: $p$ = 2.0992*10$^{-6}$; chi-square goodness-of-fit test). Compared to A17, DSI is significantly stronger in the output layers of A18 ($p$ = 0.0024; Wilcoxon rank-sum test), but there is no significant difference in the input layers ($p$ = 0.8742; Wilcoxon rank-sum test). **(B, E, and H)** Population-averaged direction tuning of response strength within the output layers (blue lines) and within the input layers (red lines). The blue and red horizontal lines represent the average responses to blank stimuli within the output layers and within the input layers, respectively. **(C, F, and I)** Population-averaged direction tuning of the response ratios between the output layers and the input layers. Data and code that support these findings are available at: https://doi.org/10.5281/zenodo.14177191.
(EPS)

## Author Contributions

**Conceptualization:** Weifeng Dai, Tian Wang, Dajun Xing.

**Data curation:** Weifeng Dai, Tian Wang, Yang Li, Yi Yang, Yange Zhang, Yujie Wu, Tingting Zhou, Dajun Xing.

**Formal analysis:** Weifeng Dai.

**Funding acquisition:** Dajun Xing.

**Investigation:** Weifeng Dai, Tian Wang, Yang Li, Yi Yang, Yange Zhang, Yujie Wu, Tingting Zhou, Dajun Xing.

**Methodology:** Weifeng Dai, Tian Wang, Yang Li, Yi Yang, Yange Zhang, Yujie Wu, Tingting Zhou, Dajun Xing.

**Project administration:** Weifeng Dai, Hongbo Yu, Liang Li, Yizheng Wang, Gang Wang, Dajun Xing.

**Resources:** Dajun Xing.

**Software:** Weifeng Dai.

**Supervision:** Dajun Xing.

**Validation:** Weifeng Dai.

**Visualization:** Weifeng Dai.

**Writing – original draft:** Weifeng Dai, Tian Wang, Dajun Xing.

**Writing – review & editing:** Weifeng Dai, Tian Wang, Hongbo Yu, Liang Li, Yizheng Wang, Gang Wang, Dajun Xing.

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
