## [Editor Report · Decision Letter 0]

11 Jun 2024

Dear Dr Xing, 

Thank you for submitting your manuscript entitled "Laminar Mechanisms for Direction Selectivity in the Primary Visual Cortex" for consideration as a Research Article by PLOS Biology.

Your manuscript has now been evaluated by the PLOS Biology editorial staff and I am writing to let you know that we would like to send your submission out for external peer review. Please note that we unfortunately have not been able to receive advice from one of our academic editors on your study and have, therefore, not yet made firm decision on whether the conceptual advance is sufficient for PLOS Biology. We will discuss this after review with one of our editorial board members and will be looking for strong reviewer support.

Once your full submission is complete, your paper will undergo a series of checks in preparation for peer review. After your manuscript has passed the checks it will be sent out for review. To provide the metadata for your submission, please Login to Editorial Manager (https://www.editorialmanager.com/pbiology) within two working days, i.e. by Jun 13 2024 11:59PM.

Kind regards,

Christian

Christian Schnell, PhD, 

Senior Editor

PLOS Biology

cschnell@plos.org

---

## [Decision Letter · Decision Letter 1]

1 Aug 2024

Dear Dr Xing,

Thank you for your patience while your manuscript "Laminar Mechanisms for Direction Selectivity in the Primary Visual Cortex" was peer-reviewed at PLOS Biology. It has now been evaluated by the PLOS Biology editors, an Academic Editor with relevant expertise, and by several independent reviewers. 

In light of the reviews, which you will find at the end of this email, we would like to invite you to revise the work to thoroughly address the reviewers' reports.

As you will see below, the reviewers are overall supportive of publication of your manuscript but they raise a number of concerns which we think need to be addressed all. In particular, Reviewer 3 asks for additional analyses and for the consideration of alternative computational models and interpretations that might account for the increased direction selectivity seen in the output responses.

Given the extent of revision needed, we cannot make a decision about publication until we have seen the revised manuscript and your response to the reviewers' comments. Your revised manuscript is likely to be sent for further evaluation by all or a subset of the reviewers.

**IMPORTANT - SUBMITTING YOUR REVISION**

*Re-submission Checklist*

*Published Peer Review*

*PLOS Data Policy*

*Blot and Gel Data Policy*

Sincerely,

Christian

Christian Schnell, PhD

Senior Editor

PLOS Biology

cschnell@plos.org

REVIEWS:

Reviewer #1: The basic orientation and direction selectivity of V1 neurons have always been one of the major focuses in vision science. In rodents the direction selectivity is initially generated in their retina. In primates and cats, the direction selectivity is firstly emerged in their V1, however, at least two major questions remain elusive regarding direction signal processing. One is the direction signal processing and transmission across hierarchical organized visual pathway such as from V1 to MT and MST in primates, and the other is the laminar processing across input and output layers within a direction selective column particularly in cat V1 and primate MT as these two areas in different species have clearly column organizations of direction selectivity. 

By taking advantages of laminar recordings with a multielectrode linear arrays (U-Probe), Dai et. al. studied direction selectivity between input and output layers precisely within a direction-selective column in cat V1. They revealed quantitively the differences of both neuronal response latency and direction selectivity across input and output layers within a single cortical column. They further constructed computational models to reveal a nonlinearity of processing explains the later but higher direction selective responses for the output layer in the preferred direction of a single cortical column. The Granger causality analysis supported a gain amplification of preferred direction selectivity caused by this nonlinearity, which most likely resulted from a combination of feedforward connections from input layer to output layer and recurrent connections within the output layer. 

Overall, this is an important and high-quality work, for the first time, Dai et. al. clearly demonstrated differential laminar processing mechanisms for direction selectivity across the input and output layers within a single cortical column in terms of processing timing and direction selectivity strength. Furthermore, the finding of stronger direction selectivity in the output layer is highly consistent with the fact that direction selectivity maps along with orientation maps are predominant features in cat V1, famously revealed by intrinsic signal optical imaging of the output layers of cat V1 something 30 years ago. These results may provide some general insights for the laminar processing of input and output layers for other non-visual sensory signal selectivity within a single column in different primary sensory cortices across species. 

However, I do have some minor issues, and also some suggestions regarding the clarity of the manuscript. 

For the title and introduction section 

1. Line 1 for the title: the title should be more specific regarding the actual laminar mechanism revealed for direction selectivity in this study. So, it would be clearer if the title to be changed as Laminar gain mechanisms for direction selectivity in the primary visual cortex. Th gain modulation mechanisms refer a more linear processing for the input layer and a more nonlinear processing for output layer in cat V1, as far as I understand from the current work. 

2. Line 49: Cat V1 traditionally was often called area 17, but not area 18. So, delete /18. 

3. Line 86: "consistent with this hypothesis" seems pointing to the above inheritance from early stages such as LGN for direction selectivity. However, the main sentence talks about the interlaminar connections from other V1 cells for the generation of the selectivity, clearly direction selectivity is not inherited from other V1 direction selective cells. This sentence is confusing and not consistent with the above inheritance hypothesis at all. 

4. Line 98 - 100. This is talking about the processing of direction signals across different hierarchical visual areas, which is another important question, in addition to the laminar processing mechanism across different cortical layers within a single cortical column. I would like to suggest the sentence of "Therefore, the manner in which direction selectivity is processed and relayed through consecutive stages of the visual hierarchy remains to be fully elucidated", to be changed to a descriptive statement of the past literatures, such as "Therefore, direction selectivity is processed and relayed through consecutive stages of the visual hierarchy". 

5. Line 107, replace "within V1 (defined here as Area 17 and 18)" with "within cat area 17 (defined here as V1)", which is more proper and precise. Remove area 18 wherever it appears in the manuscript including figure legends. Cat area 18 is often regarded as the secondary visual cortex, similar to monkey V2. 

6. Line 111, replace "hierarchical" with laminar, which is more precise and proper. Within a cortical area across different processing layers, it is more proper to use laminar, as laminar processing mechanism is the correct and precise term to use here. 

7. Line 120, this discovery of gain mechanism could be stated clearly in the title as one sentence summary title. 

For the result section 

The neuronal response latency provides a vital information for laminar processing across different cortical layers. One advantage of linear recording array across cortical layers is to reveal directly latency differences of distinct processing particularly between input and output layers. Therefore, it would be much better to combine Figure 1AB with Figure S2 as a new Figure 1, to address the distinct latency difference across input and output layers. Further differences for direction selectivity processing across input and output layers shall follow the new Figure 1 as Figure 2, and etc, thereafter. 

8. Line 281 - 282, The sub-title of "A direction-turn gain mechanism for interlaminar signal transmission leads to enhanced direction selectivity in the V1 output layer ". 

The direction-turn gain mechanism refers to selective response amplification in the preferred direction while less active in the non-preferred direction. This gain increase for preferred direction in the output layer could well be caused by excitation inputs while the gain decrease (fewer active responses) caused by more inhibitory inputs to non-preferred direction. However, the Granger causality analysis revealed the direction-turn gain likely stems from a combination of feedforward connections from input layer to the output layer, together with the recurrent connections within the output layer. The authors need to work out or at least to discuss whether the feedforward connections are more likely to be excitatory while the recurrent inputs are inhibitory. Otherwise, the suggestion from Granger causality analysis is less specific. 

9. Line 385 (Figure 6C)：The scale bar of the heatmap in Figure 6C is inconsistent with that in Figure 6AB, causing the same color in the Figure 6C represents different normalized GC values as in the Figure6AB. It is very hard to compare intuitively and fairly. The higher the normalized GC values, the lower the corresponding p-value. For example, the color of the area representing L4→L5 in the heatmap of Figure 6C shows more yellow than that in the Figure 6A, but the p-value of L4→L5 in the Figure 6C is higher than 0.05 while that in the Figure 6A is lower than 0.001. 

For the discussion section

10. Line 421: Replace "Hierarchical processing of direction selectivity in V1" with "Laminar processing of direction selectivity in V1". This is because again that Laminar processing is more precise term to use than hierarchical processing in the current study, the later often refers processing across different visual areas rather than different cortical layers. Both Laminar and hierarchical processing are equally important in the visual brain. 

11. Line 430 - 433: delete" Whether the strategy observed within a single cortical column of cat V1 is also utilized across two consecutive stages from different cortical areas (for example, from V1 to MT and from V1 to V2) remains an open question for future research." This is a less relevant question to the current study so no need to be commented here. Instead, it would be more relevant talking about there is a direction map in cat V1, but not in monkey V1, so that cat V1 provides an idea model to study laminar processing of direction selectivity across input and output layers (with citation for the direction maps recorded with intrinsic signal optical imaging). It would be more relevant to talk about cat area 18, which often regarded as equivalent to monkey V2 thick stripes. 

12. Line 515: It is difficult to say whether the use of anesthetized animals is a disadvantage for studying basic visual features such as orientation and direction selectivity. It will be definitely the case for studying any behavior visual task in anesthetized animals. 

13. Line 527 - 529: The reviewer does think these are advantages, not the limitation, as it is only fair for a direct comparison of direction selectivity between input and output layers, by using drifting gratings with fixed stimulus properties such as contrast, size, spatial frequency, and temporal frequency, with only the direction varying. Otherwise, it is hard and unfair to do direct comparison if stimuli vary with more parameters. 

14. Finally, as simple and complex cells are famously found in the input layer (more simple cells) and the output layer (more complex cells in layer 2/3) in cat V1 more than 50 years ago, how these two cell types may contribute differently to the direction-turn gain mechanism. This needs to be at least discussed, in addition to excitatory and inhibitory connections, which could link directly with the direction-turn gain mechanisms (linear and non-linearity). 

Reviewer #2: 

In this article, Dai and colleagues examined laminar processing of visual information with regard to the direction of stimulus movement in cat primary visual cortex. Using the multielectrode linear array, they recorded the multiunit spiking activity and quantified the strength of directional selectivity across the depth of a cortical column. The authors found that the directional selectivity strengths of input and output layers are correlated, and that directional selectivity is amplified in the output layers compared to the input layers. The authors discuss a possible gain mechanism for the signal transmission from input layers to output layers that leads to an enhanced directional selectivity in the output layers. Using computational modeling, the authors conclude that the amplification derives from the interplay between the feedforward interlaminar connections and the recurrent intralaminar connections. 

Apart from a few minor concerns, the results of this study seem solid, the experimental work and the analysis are carefully performed, and the manuscript is well written. I would like to propose the following points for better understanding of the results. 

In the abstract, the authors state that they studied "the V1 input layer and the V1 output layer" However, only three pages later, we learn that "input layer" includes layers 4 and 6, and that "output layer" refers to the combined layers 2/3 and 5. It is confusing and might be clearer to pluralize the word "layer" or use the more established term of input/output stage.

What are the orientation selective indeces for the directional selective MUAs identified in this study? I find it interesting that what looks like an orientation column in the input layers becomes a direction column at the later stage of processing. There could be two explanations for the low DSI values. Either the neuronal activity is selectively tuned to the oriented stimuli showing nearly equal responses to the two opposing directions, or it is broadly tuned and responds to all directions. It would be helpful to add a OSI distribution across the cortical depth. 

In line 435, the sentence "Based on a database of 899 cells, Kim et al. characterized the distribution of DSI across different layers of cat V1 (Kim & Freeman, 2016)" appears to be an incomplete statement and is not tied to the discussion. It would be interesting to see the authors discuss especially the difference in L6 DSI distribution.

There was some promise made in the manuscript, and the whole paragraph written in the Methods, about identifiyng the single unit recordings based on there waveform, but these interesting data have not been analyzed and interpreted in the results or the discussion.

Lines 696-700 in Figure S6A-E should be S3A-E.

In Figure S3B, the caption inside the box should be L2/3 instead of L23.

Reviewer #3 (Jose Manuel Alonso): This is an interesting paper that advances our understanding of how direction of motion is processed in primary visual cortex. The authors performed careful simultaneous measurements of direction selectivity at different layers of the cat visual cortex with multielectrode probes and, by doing so, they provide the first strong evidence that cortical direction selectivity increases from the input to the output layers of visual cortex. The data is of high quality and the authors convincingly demonstrate that their probes were carefully aligned with the perpendicular axis of the cortex (i.e. they recorded from neurons sharing the same retinotopy, orientation, and direction preference), which is an important requirement to obtain accurate input/output comparisons. The comments below are all relatively minor.

Main comments

1) The authors motivate the study by saying that 'It is believed that direction selectivity in later stages substantially inherits characteristics from earlier stages'. This statement can be very confusing to the reader because the terms 'later' and 'earlier stages' are too broad. For example, if 'earlier stages' refers to retina, the authors would be ignoring an extensive literature in carnivores and primates (starting with Hubel and Wiesel), which supports a cortical origin of direction selectivity. Whereas both cats and primates are likely to have a small subpopulation of retinal ganglion cells that are direction selective, few would argue that cortical direction selectivity is inherited from this small subpopulation in carnivores and primates. Consider adding the term 'cortical' after 'later' and 'earlier' in this sentence ('It is believed that direction selectivity in later cortical stages substantially inherits characteristics from earlier cortical stages'). Without 'cortical', this sentence implies that direction selectivity in visual cortex is inherited from direction selectivity in retina, which is not what was proposed by Adelson & Bergen, 1985 and many other studies in carnivores and primates. The authors can still argue that some cortical direction selectivity in rodents and lagomorphs is likely to be inherited from the retina, but there is no evidence suggesting that this is also the case for carnivores and primates (e.g. direction selective retinal ganglion cells in carnivores and primates could be part of the Koniocellular pathway that projects directly from the thalamus to area MT and not V1).

2) A main conclusion from the paper is that responses to the preferred direction are selectively amplified from the input to the output layers of the cortex. This conclusion is consistent with model II, but there are other alternative conclusions/interpretations. For example, based on the average illustrated in Figure 3A, it could be argued that the response to the preferred direction is preserved from input to output layers of V1, but the responses to all other non-preferred directions are selectively suppressed. Alternatively, and based on the distribution of response ratio for preferred direction illustrated in Figure 3C-D, it could be argued that the response to the preferred direction is enhanced in some cortical columns but not others, whereas the response to the blank is suppressed in nearly all cortical columns measured. Consistent with this argument, the output layers could receive global suppression for all directions and selective enhancement for the preferred direction. The authors may want to simulate some of these mechanisms to investigate whether their model II can be more accurate if it incorporates both intracortical excitation and inhibition.

3) The authors may want to report the correlation coefficients and correlation significance in the three panels of Figure 3D. There is clearly no correlation between response ratios to blank and null direction. However, there may be a significant correlation between response ratios to blank and preferred direction and there is definitely a strong correlation between response ratios to null and preferred direction. Also, by challenging the simulations to replicate these response-ratio correlations, the authors may make the model more realistic. For example, Figure 5C demonstrates that the model can replicate some of the measurements from Figure 3D. However, Figure 5C appears to reveal a strong correlation between the ratios of blank and nullDir that does not match the measurements from Figure 3D. Is this because they are plotting the gain of the ratios instead of the stimulated output responses (gain + nonlinearity) or is it because they are not using global suppression separately from gain? Again, testing a few variations of model II may help to identify the one that best matches the measurements from Figure 3D.

4) The classification of recordings into input and output layers is a reasonable approach to increase the statistical power of the dataset. However, if the authors have enough statistical power, it would be helpful to replicate Figure 3 in a supplementary figure just for comparisons between layer 4 and layers 2+3. This comparison is particularly important because layer 4 provides strong monosynaptic input to layers 2+3. However, the connections linking other input and output layers are much weaker (e.g. layer 6 to layer 2+3 or layer 4 to layer 5). The authors can ignore this comment if the sample size is not enough to restrict the comparisons to layer 4 and layers 2+3. 

Other minor comments

-Line 197. 'We observed that for most neuronal pairs within a cortical column, the preferred

direction remained largely unchanged, irrespective of whether the neurons were within

the same layer or across different layers'. They should mention here that this finding is consistent with a columnar organization of direction preference previously demonstrated in carnivores (e.g. imaging in cat: Shmuel and Grinvald, 1996; imaging in ferret: Weliky et al., 1996; multielectrode recordings in cat: Kremkow et al., 2016).

-Line 237. 'suggesting that neuronal activity in the output layer is inherently weaker than that in the input layer.' Perhaps, briefly mention the possibility that general anesthesia (propofol) may affect the differences in response strength between layers. If there is some evidence from awake animals that responses are weaker in output than input layers, they can use this evidence to argue that anesthesia is unlikely to affect the response differences across layers. 

-Line 275. 'the preservation of neuronal responses in the preferred direction in the output layer suggests that there is selective amplification of the neuronal response from the input layer to the output layer in the preferred direction.' An alternative interpretation is that the output layers suppress responses to all directions (untuned intracortical inhibition) while selectively increasing the response to the preferred direction (tuned intracortical excitation). This mechanism shares certain resemblance with the effects of alertness on directional selective cells in visual thalamus (see Figures 5 and 6 in Hei et al., 2014 from Swadlow's lab). 

-Line 439. 'Within a cortical column, we observed that a minority of the neuronal pairs demonstrated a shift in preferred directions of approximately 180 degrees (Figure 2A), corroborating earlier research (Berman et al., 1987).' These direction fractures (180 degrees change in direction preference within neighboring cortical regions) are consistent with a columnar organization of direction preference (see Shmuel and Grinvald, 1996; Weliky et al., 1996; Kremkow et al., 2016). Notice that the existence of this rapid 180 degrees shift in direction preference would be expected if the multielectrode probe is located at the border of two direction-preference columns. Similar rapid shifts have been observed with multielectrode probes inserted tangentially within cat visual cortex (Kremkow et al., 2016, figure 4g-h). Moreover, some neurons at the border of direction columns also have opposite direction preferences for the two eyes (e.g. Olianezhad et al., 2024).

-Line 461. 'Like the neural circuits responsible for the generation of direction selectivity (Mauss et al., 2017; H. H. Yang & Clandinin, 2018), the direction-tuned gain that we observed could result from increased excitation in the preferred direction (Reichardt, 1961), increased inhibition in the null direction (Barlow & Levick, 1965), or both of these effects.' This interpretation should be probably mentioned earlier in the paper, when presenting the model in the results section. 

-Line 467. 'In the V1 of cats (Nishiyama et al., 2019; Ohki et al., 2005; N. Swindale et al., 1987)

 and ferrets (Li et al., 2006)…, neurons are organized into two-dimensional direction maps'. The references here should include Shmuel and Grinvald, 1996 and Kremkow et al., 2016 for cats, and Weliky et al., 1996 for ferrets. 

- The legend of Figure 1 does not explain whether the illustrated dot in panel F represents the mean or median.

- Figures 3A-B and 4 show three curves for each color without explaining whether they represent mean ± SD in the legend. It would be clearer to use a single curve with a shaded area to illustrate mean ± SD.

- Figure 6 would facilitate the comparison of the three panels (A, B, and C) if the color bars have the same range.

---

## [Decision Letter · Decision Letter 2]

13 Nov 2024

Dear Dr Xing,

Thank you for your patience while we considered your revised manuscript "Laminar Gain Mechanisms for Direction Selectivity in the Early Visual Cortices" for publication as a Research Article at PLOS Biology. This revised version of your manuscript has been evaluated by the PLOS Biology editors, the Academic Editor and two of the original reviewers.

Based on the reviews and on our Academic Editor's assessment of your revision, we are likely to accept this manuscript for publication, provided you satisfactorily address the following data and other policy-related requests.

* We would like to suggest a different title to improve readability and accessibility for your broad audience: "Cortical direction selectivity increases from the input to the output layers of visual cortex"

* DATA POLICY:

Regardless of the method selected, please ensure that you provide the individual numerical values that underlie the summary data displayed in the following figure panels as they are essential for readers to assess your analysis and to reproduce it: 1E, 2CD, 3, 4CD, 5BCDEFG, 6BC

* CODE POLICY

We expect to receive your revised manuscript within two weeks. 

*Published Peer Review History*

*Press*

Sincerely,

Christian

Christian Schnell, PhD

Senior Editor

cschnell@plos.org

PLOS Biology

Reviewer remarks:

Reviewer #1: The authors have done a great job to revise their manuscript, and the revised manuscript is not only significantly improved, but also much clearer. This work stands with clear identification of the intracortical mechanisms between input and output layers responsible for different strengths of direction selectivity within a cortical functional domain. I am very pleased with this work and support it for publication as it is. 

Reviewer #3 (Jose Manuel Alonso): The authors have fully addressed all my comments. This is an important paper on the mechanisms of direction selectivity in visual cortex. The data are of high quality and the additional analyses in the revised paper reinforce the main conclusions.

---

## [Editor Report · Decision Letter 3]

21 Nov 2024

Dear Dr Xing,

Thank you for the submission of your revised Research Article "Cortical direction selectivity increases from the input to the output layers of visual cortex" for publication in PLOS Biology. On behalf of my colleagues and the Academic Editor, Frank Tong, I am pleased to say that we can in principle accept your manuscript for publication, provided you address any remaining formatting and reporting issues. These will be detailed in an email you should receive within 2-3 business days from our colleagues in the journal operations team; no action is required from you until then. Please note that we will not be able to formally accept your manuscript and schedule it for publication until you have completed any requested changes.

PRESS

Sincerely, 

Christian

Christian Schnell, PhD

Senior Editor

PLOS Biology

cschnell@plos.org